# A Fresh Perspective on Cyanobacterial Paralytic Shellfish Poisoning Toxins: History, Methodology, and Toxicology

**DOI:** 10.3390/md23070271

**Published:** 2025-06-27

**Authors:** Zacharias J. Smith, Kandis M. Arlinghaus, Gregory L. Boyer, Cathleen J. Hapeman

**Affiliations:** 1Hydrology and Remote Sensing Laboratory, Agricultural Research Service, US Department of Agriculture, 10300 Baltimore Ave, Beltsville, MD 20705, USA; hapeman@usda.gov; 2Oak Ridge Institute for Science and Education (ORISE), National Ocean Service, National Oceanic and Atmospheric Administration, 331 Fort Johnson Road, Charleston, SC 29412, USA; 3Department of Environmental Biology, College of Environmental Science and Forestry, State University of New York, 1 Forestry Drive, Syracuse, NY 13210, USA; glboyer@esf.edu

**Keywords:** paralytic shellfish toxins, saxitoxins, paralytic shellfish poisoning, cyanotoxins, algal toxins, cyanobacteria, harmful algal blooms, HABs

## Abstract

Paralytic shellfish poisoning toxins (PSPTs) are a class of neurotoxins most known for causing illness from consuming contaminated shellfish. These toxins are also present in freshwater systems with the concern that they contaminate drinking and recreational waters. This review provides (1) a complete list of the 84+ known PSPTs and important chemical features; (2) a complete list of all environmental freshwater PSPT detections; (3) an outline of the certified PSPT methods and their inherent weaknesses; and (4) a discussion of PSPT toxicology, the weaknesses in existing data, and existing freshwater regulatory limits. We show ample evidence of production of freshwater PSPTs by cyanobacteria worldwide, but data and method uncertainties limit a proper risk assessment. One impediment is the poor understanding of freshwater PSPT profiles and lack of commercially available standards needed to identify and quantify freshwater PSPTs. Further constraints are the limitations of toxicological data derived from human and animal model exposures. Unassessed mouse toxicity data from 1978 allowed us to calculate and propose toxicity equivalency factors (TEF) for 11-hydroxysaxitoxin (11-OH STX; M2) and 11-OH dcSTX (dcM2). TEFs for the 11-OH STX epimers were calculated to be 0.4 and 0.6 for 11α-OH STX (M2α) and 11β-OH STX (M2β), while we estimate that TEFs for 11α-OH dcSTX (dcM2α) and 11β-OH dcSTX (dcM2β) congeners would be 0.16 and 0.23, respectively. Future needs for freshwater PSPTs include increasing the number of reference materials for environmental detection and toxicity evaluation, developing a better understanding of PSPT profiles and important environmental drivers, incorporating safety factors into exposure guidelines, and evaluating the accuracy of the established no-observed-adverse-effect level.

## 1. Paralytic Shellfish Poisoning Toxins

Paralytic shellfish poisoning toxins (PSPTs) are a group of neurotoxins produced by certain eukaryotic algae first reported by Meyer et al. [1] and later discovered to also be produced by cyanobacteria [2]. This class of neurotoxins blocks the voltage-gated sodium channels responsible for the signal transduction of higher organisms [3]. This neuronal signaling inhibition can cause paralytic shellfish poisoning (PSP), a syndrome with symptoms including numbness of extremities, vomiting, and diarrhea [1] that, in extreme cases, has been linked to the deaths of humans and animals worldwide [4,5,6,7]. The incidence of PSP syndrome, and associated deaths, has greatly decreased following international implementation of PSPT regulations and proliferation of food supply PSPT monitoring. While the presence of PSPTs in marine systems was established in the scientific literature as early as 1928 [1] and reported through oral tradition centuries earlier [8], it took until 1968 for the first scientific report of saxitoxin (STX) in a freshwater cyanobacterium [2]. In the decades following, there has been a steady increase in reports of freshwater PSPTs and a growing recognition of the potential human health impacts from these toxins. Importantly, the routes of exposure to PSPTs in freshwater environments differs from those in coastal marine systems and therefore poses new scientific and public health questions that need to be addressed. There is significant uncertainty for human health risks associated with these toxins in freshwater systems as the regulations used in marine food safety cannot be directly translated to an instance of PSP from a freshwater source.

In recent years, PSPTs have been regularly referred to as “Saxitoxins” in freshwater environments, replacing the original name given to this class of compounds for freshwater only. We have elected to use the name paralytic shellfish poisoning toxins (PSPTs) in this review because “Paralytic Shellfish Poisoning” refers to the syndrome experienced by humans where the toxins of this class are the cause of this syndrome. We believe referring to the syndrome will help clarify confusion when using “paralytic shellfish toxins” (PSTs), an acronym commonly used in papers studying PSPTs in marine systems. While “Saxitoxins” is not inherently wrong, it is applying a new name to a long-established class of compounds, which adds unreasonable confusion to the scientific record. This same class of compounds is not referred to by the same term in marine environments. The use of Saxitoxin has also caused scientists to assume that saxitoxin, the toxin whose structure was first elucidated, to be the critical component of freshwater PSPT mixtures rather than a component of a complex mixture of toxins produced by multiple algal and cyanobacterial genera. Saxitoxin has not been established as the primary component of freshwater PSPTs, and therefore, we urge future research to avoid the term “Saxitoxins” to avoid inconsistency with more than nine decades of historical literature. While the name change was popularized because the word “Shellfish” could be confusing when discussing these toxins in the context of freshwater systems, we believe that referring to “Paralytic Shellfish Poisoning” to describe the symptoms of intoxication, rather than the source or the toxins, will clarify for stakeholders. This nomenclature retains chemical naming integrity for the scientific record and reduces the singular focus on saxitoxin as “the” PSPT of concern.

### Understanding Freshwater Paralytic Shellfish Poisoning Toxins: How Do Marine PSPTs Compare to Their Freshwater Counterparts?

PSPTs are commonly reported in marine systems [7], and the vast majority of incidents of human poisoning originate from contaminated food products. There has been a growing recognition of human health concerns from freshwater PSPTs in recent decades, but the limited data regarding the distribution, ecology, biology, and toxicity of freshwater PSPTs has made it difficult to quantify risk. Studies common to other freshwater cyanotoxins—such as those evaluating toxin production by cyanobacteria in vitro [9,10,11], environmental controls on toxin production in situ [12,13], and evaluation and development of analytical methodologies [14]—are limited for freshwater PSPTs. These data limitations extend to many other important facets of marine and freshwater PSPT research, including routes of exposure, types of PSPTs present, toxicity assessments, and analytical methodology. Recognition of the differences is essential so as to not misapply the current marine PSPT paradigm to freshwater systems and expose the public to excess risk.

Routes of PSPT exposure in marine environments are different than those typically expected to occur in freshwater systems. The primary route of exposure is through consuming contaminated shellfish that have fed on toxin-containing dinoflagellates and accumulated PSPTs in their tissues. This is typically short term and acute exposure, and toxic potential is contingent on the presence of food in the gut. In contrast, freshwater PSPT intake can be broadly understood to occur (1) by drinking water that contains planktonic cyanobacteria and/or dissolved toxins that were released into the water column; (2) through shoreline scums that can result in recreational exposure via oral, dermal, or inhalation routes; or (3) from benthic cyanobacteria that can be inadvertently consumed by pets or young children through recreational contact or from continuous release of PSPTs from mats that are released into the water column. Differences in production, distribution, and bloom dynamics between benthic and planktonic cyanobacterial producers of PSPTs create unique challenges for detection and public health monitoring. Planktonic and benthic cyanobacteria require different approaches to sampling, extraction, measurement, and interpretation of results.

The comparison of freshwater and marine cyanobacteria is further complicated by the composition of their respective PSPT congeners, referred to as the PSPT profile. This is exemplified by the discovery and isolation of novel PSPTs, the *Lyngbya wollei* toxins (LWTXs) [15], from *Microseira wollei* (basionym *Lyngbya wollei*) [16]. There are many “new” PSPTs, in addition to the LWTXs, that have been structurally elucidated and identified in freshwater and marine systems but have limited or no toxicological data. The data we do have for these toxins are not always from a biologically relevant route of exposure (e.g., intraperitoneal injection), which can have a highly significant impact on the results, as demonstrated by the recently evaluated toxicity of dihydro-anatoxin [17] and of various PSPTs [18].

Choosing the appropriate analytical tool for the measurement of freshwater PSPTs is critical for properly evaluating their occurrence and potential health risks. Current analytical methods are geared towards the detection of STX, neosaxitoxin (NEO), and some other closely related and mono- or bi-sulfated congeners such as the gonyautoxins (GTXs) and C-toxins, respectively. These techniques are not necessarily as effective, or are completely ineffective, for congeners with additional and/or different structural modifications. Understanding the fundamental processes for the operation of these methods is important to minimize errors. Neglect of the fundamentals could, and has, resulted in defective research studies and human health evaluations based on misguided or erroneous assumptions and practices.

Each additional layer of uncertainty makes a freshwater PSPT risk assessment more difficult. Due to the interdependence of the facets we use to understand and calculate an exposure risk, uncertainty associated with each facet introduces a new concern about whether there are unrecognized human health threats from PSPTs. In the following sections, we summarize the state of the science for the crucial aspects of freshwater PSPTs with the goal to aid the scientific community in addressing these gaps. The sections are as follows: (2) a table of currently known PSPTs (and some unknowns) that has not been fully summarized in the existing literature; (3) a summary of the literature identifying PSPTs in natural systems with a focus on the types of toxins, their putative producers, and the methods used for analysis; (4) an overview of the primary certified methods for PSPT and ELISA analyses, including background on the development and principles of the techniques; (5) a discussion on the application of these PSPT methods to freshwater matrices, including how to select an appropriate methodological approach and evaluate the results within the limitations of each method; (6) a summary of PSPT toxicology and how it applies today, including the origins of the current reference dose and derivation of the 80 µg STX/100 g shellfish regulatory limit and how it has been adapted to calculate freshwater exposure and consumption thresholds; (7) recommendations for quantification and reporting of environmental PSPTs and calculating a toxicity within the context of the large risk uncertainties from exposure to these toxins.

## 2. Structures of Paralytic Shellfish Poisoning Toxins

### 2.1. PSPT Structures

We have assembled a comprehensive list of PSPTs reported in the literature and shown in Figure 1 with associated structural information in Table 1. Our review found multiple prominent papers on PSPTs have incorrectly reported structures and stereochemistry [19], and we have aimed to rectify those errors. There are at least 84 structurally elucidated PSPTs, but there may be over 100+ for reasons described later in this section. When PSPTs were first isolated from shellfish and dinoflagellates, the three fractions were represented by A, B, and C chemical groups representing non-sulfated, mono-sulfated, and bi-sulfated congeners, respectively. While this naming scheme remains in place for the C-toxins, A-toxins have largely been supplanted by modern compound names (e.g., STX, NEO), and B-toxins have been (mostly) classified as the gonyautoxins (GTXs). Several of the PSPTs in Figure 1 were detected in natural systems with their structures tentatively identified through mass spectrometry (shown in grey). Due to the inherent limitations of mass spectrometry in structural identification, readers should be cautious when studying these highlighted compounds as absolute chemical configuration may change as additional structural information is collected.

There have been several PSPT terminologies used in the literature and in common parlance over the past several decades. This paper seeks to unify the congener descriptors based on the most common and structurally informative formats identified in the literature. We have opted to describe PSPTs with the *presence* of a hydroxyl group at C-11 to be 11-OH, the *absence* of a hydroxyl group at C-12 as 12α/β-do-, and the *presence* of a hydrogen at C-13, representing decarbamoyloxy-, as doSTX.

We have elected to name each derivative in the “M-toxin” class by the original “11-hydroxy…” terminology used in the 1970s–1990s to clarify which congener(s) are being discussed based on the substituents in the name [20,21,22]. This homogenizes the terminology used to describe C-12-modified toxins with the C-11-modified M-toxin group. The unified terminology makes it easier for readers to understand what congener is being discussed based on the changes at C-11 and C-12 as these compounds are determined based on the hydroxyl groups added or removed from each carbon. We kept the original name given to the M-toxins where the C-4 and N-9 bond is broken (M5,6,11,12 and dcM6,12) (Figure 1) [23].

There are a number of functional group changes at specific R groups that have not been identified in combination with each other in environmental samples. It is plausible that these compounds exist but have yet to be detected or elucidated, either due to low abundance or the incapability of current methodologies in regular employ to separate and/or detect these toxins. Some of these congeners are structurally similar to existing PSPTs, for example, doNEO and 12β-do-NEO. As there are many well-known sulfated STX and NEO analogs, such as GTX1-4, it is therefore possible that these two doSTX congeners also have corresponding NEO analogs. A non-exhaustive list of unidentified compounds in this “class” includes: doGTX1,2,3,4, 12β-do-GTX2, 12β-do-dcGTX1,2,3,4, and 12,12-dido-doNEO. It may be useful to include the molecular weights of some of these compounds in mass spectrometry (MS)-based methodologies, as many of these purported congeners could be difficult to detect (or separate) with other primary PSPT analytical techniques, as described later.

Many new PSPTs have been characterized in the last two decades, and additional producers of “unusual” PSPTs have been characterized. The first of the M-toxins 1–5 were described by Dell’Aversano et al. in 2008 [24] with the structure of one PSPT (M5) not fully characterized. The structure of M5, as well as the structures of 18 additional M-toxins, were reported by Quilliam et al. in 2017 in a conference paper [23]. The presence of the α form of several M-toxins could not be absolutely identified due to limitations on separation, but the tautomerization of GTX2,3 also likely occurs in these epimers, and so it is likely both isomeric forms exist in nature [23]. Two additional hemiaminal M-toxin variants, where N-9 is bound covalently to C-12 and termed M5-HA and M6-HA, were reported by Numano et al. in 2021 and were theorized to be scallop metabolites (Figure 1) [25]. Most environmental M-toxin detections have so far originated from shellfish tissues, but their detection in cyanobacterial cultures [26] and some environmental samples [4] suggests that some of the M-toxins are biosynthetic products or intermediates and not only metabolites and/or degradation products as was originally proposed [24]. At least one M-toxin congener, 11-OH dcSTX (dcM2), was detected and confirmed by LC-MS/MS in *Microseira wollei*, analyzed by the authors and collaborators [4].

Many PSPTs reduced at C-12 have been detected as algal and cyanobacterial PSPT congeners in recent years. LWTX4 (12β-do-dcSTX) was first identified in *Microseira* (basionym *Lyngbya*) *wollei* from Alabama [15] but was recently detected as an intermediate in the dinoflagellate genus *Alexandrium* [27]. This compound was also detected in every cyanobacterial strain examined by D’Agostino et al. in 2019, including strains of the PSPT producing genera *Heteroscytonema* (basionym *Scytonema)* [28,29], *Microseira*, *Raphidiopsis* (basionym *Cylindrospermopsis*) [30], *Aphanizomenon*, and *Dolichospermum* (basionym *Anabaena*) [26]. Another reduced 12-C derivative, 12β-do-GTX5, was detected by Akamatsu et al. in both the dinoflagellate *Alexandrium* and the cyanobacterium *Dolichospermum* [31], while 12β-do-GTX3 was detected in *Anabaena* (basionym *Dolichospermum*) *circinalus* [32]. 12,12-dido-doSTX and 12β-do-doSTX were synthesized and detected in *Alexandrium* and *Dolichospermum* but no α C-12 variants were reported [33]. Hakamada et al., 2024 [33] stated they did not identify α isomers in agreement with Akamatsu et al., 2022 [31] and reported that there have been no natural identifications of 12α-do PSPT derivatives, while analysis of cyanobacterial strains by D’Agostino et al., 2019 [26] identified two sets of α isomers in the majority of the strains that they investigated. Further research is required to understand the discrepancies regarding the existence of both α and β C-12 stereoisomers in nature.

Several unique PSPTs have been detected in higher organisms: the 11-saxitoxinethanoic acid (SEA) in crab [34], a methylated carbamoyl derivative STX-uk from pufferfish [35], and zetekitoxin AB from golden frogs [36]. The formation of these congeners likely varies between each organism, but it is possible some are metabolites formed in the organisms, as these PSPTs have not yet been reported in known PSPT producing dinoflagellates or cyanobacteria.

**Table 1 marinedrugs-23-00271-t001:** Structural information of PSPT congeners including the source of the structural detection and/or elucidation. Molecular weights and formulae for each compound are at neutral pH, with both guanidinium groups protonated and any sulfates deprotonated. In addition to the PSPTs shown here, there are three additional PSPT congeners reported in higher organisms (details in text). STX, saxitoxin class; GTX, gonyautoxin class; LWTX, *Lyngbya wollei* toxin class; dc, decarbamoyl; do, decarbamoyloxy at 13-C; 11-OH, hydroxyl at C-11; 12-do, deoxy at C-12.

	Name	Elemental Formula	M.W.	Exact Mass	Original Elucidation
1.	Neosaxitoxin (NEO)	C_10_H_19_N_7_O_5_^2+^	317.30	317.14367	[37,38]
2.	Saxitoxin (STX) *	C_10_H_19_N_7_O_4_^2+^	301.30	301.14875	[39,40]
3.	12,12-dido-dcSTX	C_9_H_18_N_6_O^2+^	226.28	226.15311	[26,33]
4.	12α-do-doSTX	C_9_H_18_N_6_O^2+^	226.28	226.15311	[26]
5.	LWTX4 (12β-do-dcSTX)	C_9_H_18_N_6_O_2_^2+^	242.28	242.14803	[15,27]
6.	12α-do-dcSTX	C_9_H_18_N_6_O_2_^2+^	242.28	242.14803	[26]
7.	12β-do-doSTX	C_9_H_18_N_6_O^2+^	226.28	226.15311	[26]
8.	12β-do-STX	C_10_H_19_N_7_O_3_^2+^	285.30	285.15384	[31]
9.	12,12-dido-doSTX	C_9_H_18_N_6_^2+^	210.28	210.15820	[33]
10.	C1	C_10_H_17_N_7_O_11_S_2_	475.41	475.04275	[41,42]
11.	C2	C_10_H_17_N_7_O_11_S_2_	475.41	475.04275	[42]
12.	C3	C_10_H_17_N_7_O_12_S_2_	491.41	491.03766	[37]
14.	C4	C_10_H_17_N_7_O_12_S_2_	491.41	491.03766	[37]
15.	GTX1	C_10_H_18_N_7_O_9_S^+^	412.36	412.08812	[21,43,44]
16.	GTX2 *	C_10_H_18_N_7_O_8_S^+^	396.36	396.09321	[20,22]
17.	GTX3 *	C_10_H_18_N_7_O_8_S^+^	396.36	396.09321	[20,22]
18.	GTX4	C_10_H_18_N_7_O_9_S^+^	412.35	412.08812	[43,45]
19.	GTX5 (B1)	C_10_H_18_N_7_O_7_S^+^	380.36	380.09829	[46]
20.	GTX6 (B2)	C_10_H_18_N_7_O_8_S^+^	396.35	396.09321	[46]
21.	dcGTX1	C_9_H_17_N_6_O_8_S^+^	369.33	369.08231	[47]
22.	dcGTX2	C_9_H_17_N_6_O_7_S^+^	353.33	353.08739	[15,48]
23.	dcGTX3	C_9_H_17_N_6_O_7_S^+^	353.33	353.08739	[15,48]
24.	dcGTX4	C_9_H_17_N_6_O_8_S^+^	369.33	369.08231	[47]
25.	dcNEO	C_9_H_18_N_6_O_4_^2+^	274.28	274.13786	[49,50]
26.	dcSTX	C_9_H_18_N_6_O_3_^2+^	258.28	258.14294	[51,52,53] **
27.	doSTX	C_9_H_18_N_6_O_2_^2+^	242.28	242.14803	[49,50]
28.	LWTX1	C_11_H_19_N_6_O_7_S^+^	379.37	379.10304	[15]
29.	LWTX2	C_11_H_19_N_6_O_8_S^+^	395.37	395.09796	[15]
30.	LWTX3	C_11_H_19_N_6_O_8_S^+^	395.37	395.09796	[15]
31.	LWTX5	C_11_H_20_N_6_O_4_^2+^	300.31	300.15351	[15]
32.	LWTX6	C_11_H_20_N_6_O_3_^2+^	284.31	284.15859	[15]
33.	11α-OH GTX5 (M1α) ^†^	C_10_H_18_N_7_O_8_S^+^	396.36	396.09321	[23]
34.	11α-OH STX (M2α) ^†^	C_10_H_19_N_7_O_5_^2+^	317.30	317.14367	[20,24]
35.	11β-OH GTX5 (M1β)	C_10_H_18_N_7_O_8_S^+^	396.36	396.09321	[24]
36.	11β-OH STX (M2β)	C_10_H_19_N_7_O_5_^2+^	317.30	317.14367	[20,24]
37.	11β-OH dcSTX (dcM2β)	C_9_H_18_N_6_O_4_^2+^	274.28	274.13786	[23,43]
38.	11α-OH dcSTX (dcM2α) ^†^	C_9_H_18_N_6_O_4_^2+^	274.28	274.13786	[23,43]
39.	11,11-OH GTX5 (M3)	C_10_H_18_N_7_O_9_S^+^	412.09	412.08812	[24]
40.	11,11-OH STX (M4)	C_10_H_19_N_7_O_7_^2+^	349.30	349.13350	[24]
41.	11,11-OH dcSTX (dcM4)	C_9_H_18_N_6_O_6_^2+^	306.28	306.12769	[23]
42.	M5	C_10_H_20_N_6_O_10_S^+^	430.37	430.09869	[23,24]
43.	M6	C_10_H_21_N_7_O_7_^2+^	351.31	351.14915	[23]
44.	dcM6	C_9_H_20_N_6_O_6_^2+^	308.29	308.14334	[23]
45.	11α-OH GTX6 (M7α) ^†^	C_10_H_18_N_7_O_9_S^+^	412.36	412.08812	[23]
46.	11β-OH GTX6 (M7β)	C_10_H_18_N_7_O_9_S^+^	412.36	412.08812	[23]
47.	11α-OH NEO (M8α) ^†^	C_10_H_19_N_7_O_6_^2+^	333.30	333.13858	[23]
48.	11β-OH NEO (M8β)	C_10_H_19_N_7_O_6_^2+^	333.30	333.13858	[23]
49.	11α-OH dcNEO (dcM8α) ^†^	C_9_H_18_N_6_O_5_^2+^	290.28	290.13277	[23]
50.	11β-OH NEO (dcM8β)	C_9_H_18_N_6_O_5_^2+^	290.28	290.13277	[23]
51.	11,11-OH GTX6 (M9)	C_10_H_18_N_7_O_10_S^+^	428.36	428.08304	[23]
52.	11,11-OH NEO (M10)	C_10_H_19_N_7_O_7_^2+^	349.30	349.13350	[23]
53.	11,11-OH dcNEO (dcM10)	C_9_H_18_N_6_O_6_^2+^	306.27	306.12769	[23]
54.	M11	C_10_H_20_N_7_O_10_S^+^	430.37	430.09869	[23]
55.	M12	C_10_H_21_N_7_O_7_^2+^	351.32	351.14915	[23]
56.	dcM12	C_9_H_20_N_6_O_6_^2+^	308.29	308.14000	[23]
57.	12β-do-GTX3	C_10_H_18_N_7_O_7_S^+^	380.36	380.09829	[20,32]
58.	12β-do-GTX2	C_10_H_18_N_7_O_7_S^+^	380.36	380.09829	[20]
59.	12β-do-GTX5	C_10_H_18_N_7_O_6_S^+^	364.36	364.10338	[31]
60.	M5-HA	C_10_H_19_N_7_O_5_^2+^	317.30	317.14367	[25]
61.	M6-HA	C_10_H_18_N_7_O_9_S^+^	412.35	412.08812	[25]
62.	12β-do-doSTX (doLWTX4)	C_9_H_18_N_6_O^2+^	226.28	226.15311	[26]
63.	12α/β-do-GTX4	C_11_H_20_N_7_O_8_S^+^	410.38	410.10886	[54]
64.	GC1	C_16_H_21_N_6_O_9_S^+^	473.44	473.10852	[55]
65.	GC2	C_16_H_21_N_6_O_9_S^+^	473.44	473.10852	[55]
66.	GC3	C_16_H_22_N_6_O_5_^2+^	378.38	378.16407	[55]
67.	GC4	C_16_H_21_N_6_O_10_S^+^	489.44	489.10344	[56]
68.	GC5	C_16_H_21_N_6_O_10_S^+^	489.44	489.10344	[56]
69.	GC6	C_16_H_22_N_6_O_6_^2+^	394.38	394.15899	[56]
70.	GC1a	C_16_H_21_N_6_O_10_S^+^	489.44	489.10344	[56]
71.	GC2a	C_16_H_21_N_6_O_10_S^+^	489.44	489.10344	[56]
72.	GC3a	C_16_H_22_N_6_O_6_^2+^	394.39	394.15899	[56]
73.	GC4a	C_16_H_21_N_6_O_11_S^+^	505.43	505.09835	[56]
74.	GC5a	C_16_H_21_N_6_O_11_S^+^	505.43	505.09835	[56]
75.	GC6a	C_16_H_22_N_6_O_7_^2+^	410.38	410.15390	[56]
76.	GC1b	C_16_H_20_N_6_O_12_S^2+^	552.49	552.05806	[56]
77.	GC2b	C_16_H_20_N_6_O_12_S_2_	552.49	552.05806	[56]
78.	GC3b	C_16_H_21_N_6_O_8_S^+^	457.44	457.11361	[56]
79.	GC4b	C_16_H_20_N_6_O_13_S_2_	568.49	568.05298	[56]
80.	GC5b	C_16_H_20_N_6_O_13_S_2_	568.49	568.05298	[56]
81.	GC6b	C_16_H_21_N_6_O_9_S^+^	473.44	473.10852	[56]

* The structures initially reported were incorrect and were later corrected by other research groups. We included the incorrect structures to retain the historical record of the original research. The correct structure of GTX2,3 was first reported by Boyer et al. in 1978 [20] and STX by Schantz et al., 1975 [40]. ** dcSTX was first isolated and labeled PBT1 (Oshima, Y.; personal communication). ^†^ Compounds were not directly observed due to difficulty in separating isomers but are believed to exist. Tautomerization between C-11 and C-12 due to a keto-enol equilibration leads to the loss of stereospecificity of GTX2,3 over time and would likely lead to the formation of both pairs of C-11 11-OH epimers (see Section 2.2 for a detailed explanation).

A number of PSPT-like compounds have been detected with incomplete structural assignments due to limitations in analytical methodology, but converging lines of evidence suggest that these compounds were truly PSPTs and not artifacts or interferences. Four PSPTs were detected in clams by Vale et al. in 2009 but were not structurally characterized and given names A-D [57]. Vale also identified 15 additional hydroxybenzoated PSPT [58] congeners in the dinoflagellate *Gymnodinium catenatum* (Figure 1, Table 1) but could not unequivocally assign structures, with some of the compounds identified containing constitutional isomers [56]. A compound was identified in Lake Baikal from scrapes of benthic algal material and identified by MALDI-MS as “doGTX2,3” as described in the text and “doGTX2,4” in a table [59]. Smith et al., 2019 [12] identified a compound in *Microseira wollei* by post-column chemical oxidation (PCOX) that had the same retention time as GTX5, had a [M + H]^+^ of 380 AMU and a MS/MS transition of 380 → 300, potentially representing the loss of a sulfate moiety. This compound eluted 0.1 min away from authentic GTX5 by HILIC UPLC chromatography and identically to authentic GTX5 in PCOX; however, the exact mass for GTX5 was not detected by high resolution mass spectrometry, nor was 12β-do-GTX2/3, which shares a 380 AMU molecular weight [12]. Several PSPTs were identified in Cayuga Lake, New York, but their structures could not be determined as the retention times for the majority of compounds did not match with PSPTs for which there were commercially available standards [4,60]. Several unknown PSPTs were identified in strains of *Raphidiopsis* (*Cylindrospermopsis*) *raciborskii* isolated from Brazil, at times in high abundance, with the [M + H]^+^ of one of the compounds reported as 426 AMU [61,62]. In 2022, Leal and Christiano reported structures and molecular weights of doGTX1 and doGTX2 in their review, but it is not clear whether these were found in nature as they were not mentioned elsewhere in their paper [63].

Strains of *Aphanizomenon* collected in New Hampshire from Kezar Lake were found to contain STX, possibly NEO, and several other unidentified neurotoxins with structural similarity to PSPTs [64]. These compounds were termed the aphantoxins 1, 2 and 3, 4 and 5 and had properties consistent with PSPTs, including neurotoxicity in bioassays, nerve and muscle blocking on frog nerve-muscle fibers, infra-red (IR) spectral similarity, capacity for florescence via chemical oxidation, and presence of guanidinium functional groups [65,66,67,68,69,70,71]. However, there were noted differences of isolated aphantoxin fractions from known PSPTs: color reaction similarities were different from the (at the time) known PSPTs, hydrogenation did not reduce or remove bioassay toxicity, TLC retention time characteristics for many of the compounds differed from reference materials, there were notable differences in IR spectral signature, attempted cleavage of a sulfate from a (at the time presumed) C-13 N-sulfocarbamoyl functional group was unsuccessful, and some of the toxins were thermally labile, unlike previously elucidated PSPTs (e.g., STX). The literature is not clear whether these compounds were ever elucidated. One compound, aphanorphine, was isolated and characterized from these strains and may have represented one of the fluorescent compounds identified [72], although this compound appears to lack the neurotoxicity found in PSPTs.

In 2000, Li et al. stated, “In a subsequent study of the toxicity of this organism, one of us (WWC) isolated several strains of toxin producing *Aphanizomenon* from a different location to the Sawyer collections in New Hampshire, in 1980” [73]. It appears that the NH1-NH5 strains collected by Wayne Carmichael were an entirely distinct population from those collected previously and therefore may have had a different toxin profile. Analyses by Mahmood and Carmichael in 1986 [64] and by D’Agostino et al. in 2019 [26] did find PSPTs other than STX and NEO but at relatively low abundance. These compounds were likely not aphantoxins, as the properties of these toxins are likely not in agreement with those reported of the Kezar Lake toxins.

It is important to note the names for the toxin fractions were inconsistent between these sets of isolations and research papers: aphantoxin 1 and 2 were NEO and STX in Mahmood and Carmichael, 1986 [64], while aphantoxin 1 in Alam et al., 1978 [67] was not STX based on their TLC analyses and more closely resembled “GTX2”.

### 2.2. Important Chemical Features of PSPTs

PSPTs are highly polar molecules that contain two guanidinium functional groups as part of the core ring system that are charged at most lake pH conditions. Since the guanidinium pKas were measured by Rogers and Rapoport to be 8.22 and 11.28, some change in ionization is likely in many bloom conditions where pH regularly exceeds 8 [74]. In positive mass spectrometric ionization, PSPTs generally only maintain a singular charge on one guanidinium group [75].

PSPTs can have sulfur containing functional groups located on the C-11 carbon as hydroxysulfate and/or an N-sulfocarbamoyl functional group on C-13 that liberates a sulfate when hydrolyzed. This latter sulfate is exceptionally easy to hydrolyze from C-toxins and GTXs 5 and 6. Heating at 100 °C for 10 min without the addition of acid was found to remove the sulfate on N-sulfocarbamoyl derivatives, while addition of low concentrations (<0.2 M) of HCl liberated ~4× more sulfate functional groups within the same time-span [76]. While the C-11 sulfate functional group could be liberated by the same reaction, the C-11 sulfate was less labile and required higher (3 M) concentrations of HCl, although such conditions can also cause degradation of the toxins. Chemical cleavage at C-11 was more effective using 10% acetylchloride or reductive cleavage with 1,4-dithiothreitol (DTT), which selectively reduces only the C-11 while maintaining the integrity of the N-sulfocarbamoyl at C-13.

PSPTs can contain a number of hydroxyl functional groups, a geminal diol at C-12 and, in NEO-type derivatives, an N-1 hydroxy functional group. Cleavage of sulfate from C-11 or the C-13 carbamoyl functional group can leave PSPT derivatives with additional hydroxyl functional groups, while some M-toxin derivatives can contain a second geminal diol at C-11. The C-12 geminal diol is important for performing chemical oxidation reactions, where reduction to a hydroxyl functional group reduces fluorescence response by ~10-fold [27].

Stereochemistry can be lost on C-11 due to an equilibration, or tautomerization, that occurs because of the presence of a geminal diol/ketone found on C-12. There exists an equilibrium of the geminal diol that has two hydroxyl groups on the same carbon to the corresponding carbonyl compound through the loss of a water molecule, representing the “keto” form of the PSPT. Electrons centered around C-11 can form a double-bond between C-11 and C-12 and the formation of a vinyl hydroxyl at C-12 representing the “enol” form. When there is an -OH or sulfate at C-11 the subsequent stereocenter is lost while the molecule is in the enol form. When the enol reverts to the keto form, the resulting isomer could be either R or S, and therefore, isomers at C-11 exist as an equilibrium mixture (for example, GTX2 and GTX3, GTX1 and GTX4, 11α-OH STX and 11β-OH STX, and C1 and C2). While reference materials are found in epimeric mixtures, biosynthesis of these isomers may be stereospecific as the epimer ratio found in dinoflagellate (*Gonyaulax excavata)* producers has been found to be significantly different than the ratio measured in scallops [77].

The functional group of C-13 (Figure 1, R_6_) can be one of several substituents, including carbamoyl (STX and NEO), an N-sulfocarbamoyl (C toxins or GTX 5 and 6), hydroxyl (dcSTX and dcNEO), acetyl (LWTXs), and hydrogen (decarbamoyloxy derivatives). To date only the acetyl group in the LWTXs has been uniquely identified in freshwater cyanobacteria. Carbamoyl functional groups can be removed by heating for 3 h at 110 °C in 7.5 N HCl with a yield of 75% [78]. A Barton-McCombie deoxygenation [79] using azobis(isobutyronitrile) and trimethyl silane can be used to synthesize 13-C deoxy derivatives from decarbamoyl derivatives as detailed in Hakamada et al., 2024 [33].

### 2.3. Abiotic and Biological Conversions of PSPTs and Relevance to Freshwaters

PSPTs are generally highly stable compounds that degrade or transform slowly (days to weeks) in conditions common to natural systems. Direct photolysis from the sun does not lead to degradation. Instead, degradation occurs through indirect (sensitized) photolysis [80]. Direct chemical transformation in ambient aquatic pHs ~6–9 or from normal water temperatures of ~0–30 °C does not lead to rapid degradation [81,82,83].

However, there are a number of relevant transformations of PSPTs where congeners can be converted chemically and enzymatically in the environment. These transformations have been studied primarily in a marine context. Bacteria associated with shellfish were able to convert: GTX1,4 into GTX2,3, dcGTX2,3, and other unidentified non-gonyautoxin products; GTX5 into an unknown; and GTX2,3 into dcGTX2,3, and unknown, and a product that may be an M-toxin derivative based on retention time [84]. There may be bacteria in freshwater environments that perform analogous chemical conversions. The N-sulfonate group found on the carbamoyl functional group is highly labile and can be chemically cleaved with heat and/or acid [76]. Although the reaction rates would likely be heavily reduced, it is possible that conditions in some microenvironments would facilitate hydrolysis. The M-toxins were originally purported to be shellfish degradation byproducts [24]. They may still be shellfish metabolites, but some are also synthesized in cyanobacterial culture [26]. Whether these are degradation or biosynthetic products in all cases, or how this varies between the different M-toxins, is currently unknown. The skeletal structure of STX was found to be modified in scallop, but whether this transformation was due to scallop or bacterial enzymes and/or metabolism was not elucidated; if the transformation is linked to enzymes produced by the scallop [25], then it is unlikely that these derivatives would be identified in freshwater systems.

Several studies have been published on the transformation of PSPTs in freshwater systems. In drinking water treatment plants, STX, GTX2,3, and C1,2 were found to interconvert through biological processes in the sand and anthracite filter beds [85]. PSPT extracts incubated in sub-surface and river waters showed slow but marked transformation of toxin, especially for the C-toxins, with the mechanism hypothesized to be driven by chemical and catalytic processes rather than biotic [81]. A freshwater mussel incubated with toxic *Dolichospermum* had moderately different ratios of PSPTs than the source culture, suggesting a combination of preferential uptake, transformation, and/or depuration of different PSPT congeners [86].

## 3. Paralytic Shellfish Poisoning Toxins in Freshwater Environments

### 3.1. Production of PSPTs by Planktonic and Benthic Cyanobacteria

Freshwater PSPT production has been established across cyanobacteria genera, including planktonic *Aphanizomenon, Dolichospermum* (basionym *Anabaena), Planktothrix, Raphidiopsis* (basionym *Cylindrospermopsis*), and the benthic cyanobacterium *Microseira* (basionym *Lyngbya)* and *Heteroscytonema crispum* (basionym *Scytonema)*. PSPTs have been more frequently reported from planktonic cyanobacteria compared to their benthic counterparts [5], likely resulting from a historical focus on visually observable surface-water blooms. Internationally infamous harmful algal bloom events include microcystin poisonings in Brazil [87], the Palm Island Mystery Disease [88], and the 2014 Lake Erie Toledo Water Crisis [89]. Famous PSPT blooms include toxin intoxication causing PSP, leading to sheep deaths in Australia [90], and PSPT-producing *Raphidiopsis* in Brazilian drinking water reservoirs [62].

The presence of benthic “blooms” has been harder to quantify, especially in lentic (still freshwater) systems. Often their presence is established only after the death(s) of animals exposed to the water or that consumed the toxic material. As researchers continue to probe deeper, the environmental health risk associated with these benthic blooms has become better recognized [91]. Such events have been described world-wide, including but not limited to New Zealand [92], France [93], Switzerland [94], and the United States [95]. Literature reports of PSPT producing *Microseira* (*Lyngbya*) *wollei* in North America have steadily increased, while a recent identification in China is the first reported detection of a LWTX producer outside of North America. The locations where PSPT producing *Microseira* have been detected are Guntersville Reservoir, Alabama, United States [15], Blue Hole Springs and Silver Glen Springs in Florida [14], Butterfield Lake in New York [12], in Canada in the St. Lawrence River near Montreal [96], in South Carolina [97], Chesapeake Bay, Maryland [98], and in China [99]. It is certain that more reports of these benthic PSPT producers will occur both in the United States and internationally in coming decades. A significant research effort in the 1990s aimed to discover whether there was production of PSPTs by heterotrophic bacteria associated with dinoflagellates; however, there was no conclusive evidence that these bacterial isolates were distinct PSPT producers independent of their dinoflagellate hosts [100,101,102,103,104]. To our knowledge, no investigations of PSPT production by heterotrophic freshwater bacteria have been performed.

The inherent distinctions between benthic and planktonic cyanobacteria, the known producers of freshwater PSPTs, lead to differences in production, concentrations, distribution, and bloom dynamics, creating unique challenges for detection and public health monitoring. For example, the LWTXs produced by benthic *Microseira wollei* have a distinct chemical structure from most other PSPTs with an acetyl group in place of the carbamoyl found on most other PSPTs. Distribution of toxins can also affect the likelihood and type of exposure as planktonically produced PSPTs are more likely to be ingested at high concentrations from water with whole-cell material, whereas benthic cyanobacteria may be more likely to involve chronic ingestion, rather than acute intake and injury, by leakage of intracellular toxin into the water column. Assessing a “benthic bloom” is far more challenging than a planktonic one, where there is no established metric to determine when a benthic bloom reaches a concerning breadth and/or density and no universal approach to assess the size(s) of freshwater benthic blooms. Planktonic and benthic cyanobacteria require different approaches to sampling, extraction, measurement, and interpretation of toxin results due to these challenges.

### 3.2. History of Detection of PSPTs in Freshwater Environments

A detailed overview of naturally occurring PSPTs in freshwater environments worldwide and the detection techniques used to determine their presence is shown in Table 2. Table 2 summarizes the literature with the following criteria: (1) PSPTs identified in natural freshwater samples of open water or benthic algae and (2) the use of scientifically sound/reputable PSPT detection methods including post-column chemical oxidation (PCOX), pre-column chemical oxidation (PreCOX), liquid chromatography with either MS or MS/MS detection (LCMS), receptor binding assay (RBA), enzyme-linked immunosorbent assay (ELISA), and the PSPT mouse bioassay (MBA), etc. Papers excluded from Table 2 included those that only utilized MBA and therefore can only conclude neurotoxicity, those with only results reporting STX gene analyses, and toxin analyses from freshwater tissue samples or large-scale PSPT analyses (see Table 3). Genera names were modified from their original sources to modern taxonomic classification in the majority of cases where it was likely that the organism(s) of interest in each individual study would be reclassified under the new names. However, where it was unclear what organisms were identified, genera and/or species names were left unmodified from the original source as we did not want to misclassify organisms and confuse the historical record.

Many of the earliest papers reporting freshwater PSPTs are based on indirect measurements from isolated cyanobacterial cultures, as noted in Table 2. While useful in identifying toxic potential of different cyanobacteria, this approach does not represent the toxin profiles that would have been measured in situ. Toxin profiles can be changed by many environmental factors [9,12], which are modified by removing strains from their natural aquatic environment. PSPT profile information is most useful when reported from environmental samples, as these more accurately represent toxin profiles within a body of water; it must be noted that toxin profiles are not expected to be static in natural populations, but measurements would have more relevance in risk assessment compared to laboratory grown cyanobacterium. Early investigation of cyanobacterial PSPTs required isolation and culture analysis as detection limits would not always have been low enough to detect environmental concentrations of PSPTs. As limits of detection and sensitivity improved over the decades, the need for isolation of cyanobacteria prior to chemical analysis became less essential for PSPT detection.

Some exceptions were made for studies reporting laboratory culture analyses only—no direct source-water samples—[67,69,105,106] (Appendix A), with discoveries that were critical for characterizing the current suite of PSPTs and/or producers. Other examples of exceptions are Lagos et al. (1999) [62] and Sant’Anna et al. (2011) [107], who identified PSPTs in cultures of the cyanobacterial taxa *Raphidiopsis* (basionym *Cylindrospermopsis*) *raciborskii* and *Microcystis aeruginosa*, respectively, which were organisms previously unreported to produce PSPT. Only two systematic evaluations of PSPT production capability and toxin profiles by multiple cyanobacterial strains have been published to our knowledge: Onodera in 1996 and his dissertation in 2000 reported PCOX analysis results for 66 cyanobacterial culture strains [105,106], while D’Agostino in 2019 reevaluated the PSPTs biosynthesized by six distinct cyanobacterial isolates [26]. Research producing new STX ELISAs, and their application in marine and freshwater environments, was common in the late 1900s due to the impact PSPTs had on food supplies [108], while at the time of publication, Gold Standard Diagnostics/Abraxis produced one of the most widely-used commercial STX ELISA kits [109]. Many papers listed have used ELISA as their primary testing tool—while useful, ELISA kits do not offer any structural information about the PSPT congeners present (see Section 5). Since concentrations measured are not linked to toxic effects, estimates of the “true” concentration of PSPTs are generally inaccurate due to the inconsistent PSPT congener cross-reactivity (see Section 7.1).

Jackim and Gentile, 1968 are often cited as being the first to describe a PSPT originating from a cyanobacterium, with characterization by chemical, chromatographic, and infrared absorption and comparison to authentic STX [2]. However, there was an earlier 1967 isolate by Sawyer et al. [70] where “the toxin” was described to have properties similar to tetrodotoxin (TTX) (full structure reported in the mid 1960s [110,111]) and “mussel poison” (STX), which was not fully characterized until seven-years after this publication in 1975 [40]. Outside of New Hampshire, cyanobacterial PSPTs were not reported again until the early 1990s when they were detected in Australia [112]. Advancements in detection techniques led by Dr. Yasukatsu Oshima in the 1980–1990s [48] led to an increasing number of PSPT detections in the 1990s and onward, which is demonstrated by a steady increase in environmental PSPT detections up to the present day (Table 2). The United States, Australia, Brazil, Portugal, and Italy stand out in PSPT monitoring and detection during the 1990s and continue to lead in regular reports of PSPTs, alongside newer reports originating from New Zealand and Greece (Table 2).

Research investigating STX ELISAs and their application in marine and freshwater environments was common in the late 1900s due to the impact PSPTs had on food supplies [108]. At the time of publication, Gold Standard Diagnostics/Abraxis produce one of the most widely-used commercial STX ELISA kits [109]. While many studies in Table 2 used ELISA as their primary testing tool, ELISA kits do not offer any structural information about the PSPT congeners present (see Section 5). Since concentrations measured are not linked to toxic effects, estimates of the “true” concentration of PSPTs are generally inaccurate due to the inconsistent PSPT congener cross-reactivity (see Section 7).

PSPT contamination of higher-order (non-cyanobacterial) organisms has been better studied in marine environments [113], but a selection of freshwater organisms have also been analyzed for PSPTs. The earliest detection of PSPTs in these organisms was in the 1990s from Southeast Asia in freshwater pufferfish [114,115] and freshwater bivalves [116] using MBA and HPLC. PSPTs in freshwater fish [117,118,119] and snail [120] samples have been reported via ELISA in North America, South America, and Europe. These papers were omitted from Table 2 and Table 3 unless paired with water samples.

**Table 2 marinedrugs-23-00271-t002:** PSPTs detected in freshwater environments. Each ELISA assays specifies the manufacturer used—if no manufacturer was mentioned by the study authors then only ELISA is listed.

Date Collected	Location (Country)	Freshwater Source	Benthic/Open	Cyanobacterium	Detections Method	PSPT Congeners	Notes	Authors
1967	New Hampshire, United States	Kezar Lake	O	*Aphanizomenon flos-aquae*	MBA, Patch-clamp	“the toxin”	Toxins later elucidated as PSPTs	Sawyer et al., 1968 [70]
1968	-	-	O	*Aphanizomenon flos-aquae*	Infrared Spectroscopy, Chemical Assay, TLC	STX	Lab culture	Jackim & Gentile, 1968 [2]
1970	New Hampshire, United States	Kezar Lake	O	*Aphanizomenon flos-aquae*	MBA, HPLC, TLC	STX, *		Alam et al., 1978 [67]
~1970s	New Hampshire, United States	Kezar Lake, North Sutton	O	*Aphanizomenon flos-aquae*	MBA, HPLC	STX(?), 3 unknowns, *	Lab culture	Sasner et al., 1981 [65]
1980	New Hampshire, United States	Pond near Durham, NH	O	*Aphanizomenon flos-aquae*	MBA, PCOX, TLC	STX, NEO	Lab Culture	Ikawa et al., 1982 [69]
1980	New Hampshire, United States	Pond near Durham, NH	O	*Aphanizomenon flos-aquae*	MBA, HPLC	STX, NEO	Lab culture	Mahmood & Carmichael, 1986 [64]
1990–1993	Australia	Murray-Darling Basin (Millbrook Reservoir)	O	*Anabaena circinalis*	MBA, Electrophysiology, HPLC, FAB-MS	STX, GTX1,2,3,4,6, C1, C2, dcGTX2(?), 3(?), NEO(?)	Natural and Cultured	Humpage et al., 1994 [112]
1991–1994	Alabama, United States	Guntersville Reservoir, Tennessee River	B	*Microseira* (*Lyngbya*) wollei	MBA, RBA, PCOX	GTX?, dcSTX?, dcGTX2,3	Natural and Cultured	Carmichael et al., 1997 [121]
1993	Alabama, United States	Guntersville Reservoir, Tennessee River	B	*Microseira* (*Lyngbya*) *wollei*	MBA, PCOX, MS, NMR	dcSTX, dcGTX2,3, LWTX1-6	Toxins in situ and culture	Onodera et al., 1997 [15]
1993–1994	Multiple	Multiple	O/B	*Dolichospermum (Anabaena) circinalis*, *Raphidiopsis (**Cylindrospermopsis) raciborskii, Lyngbya (Microseira) wollei*	PCOX, MS	Multiple	Lab culture, see Appendix A	Onodera et al., 1996, 2000 [105,106]
1994	Australia	Dam near Forbes, Central New South Wales	O	*Dolichospermum* (*Anabaena*) *circinalus*	MBA, PCOX	STX, C1,2, dcGTX2,3, GTX2,3,5, dcSTX	Sheep mortality	Negri et al., 1995 [90]
1994 & 1996	Brazil	Amparo City and Billings Reservoirs, São Paulo	O	*Raphidiopsis* (*Cylindrospermopsis*) *raciborskii*	MBA, PCOX, LC-MS/MS	STX, NEO, GTX2,3, *	Toxin only detected in culture	Lagos et al., 1999 [62]
1996	Portugal	Crestuma-Lever Reservoir, Douro River	O	*Aphanizomenon flos-aquae*	MBA, PCOX	STX, NEO, dcSTX, GTX1,2,3,4	Natural and Cultured	Ferreire et al., 2001 [122]
1996	Portugal	Montargil Reservoir	O	*Aphanizomenon flos-aquae*	MBA, PCOX, LC-MS/MS	STX, NEO, dcSTX, GTX5,6	Natural and Cultured	Pereira et al., 2000 [123]
1996 ^†^	Australia	Burrinjuck Dam, New South Wales	O	*Dolichospermum* (*Anabaena*) *circinalis*	PCOX	C1,2, dcGTX2,3, GTX2,3	Stability over 90 days	Jones & Negri, 1997 [81]
1997	Italy	Lake Varese	O	*Planktothrix agardhii*	preCOX and PCOX, LC-MS/MS, Patch-Clamp	STX	Natural and Cultured	Pomati et al., 2000 [124]
1997	Italy	Lake Varese	O	*Oscillatoria*, *Aphanizomenon*, *Anabaena* *	preCOX, Patch-Clamp	STX	Natural and Cultured	Giovannardi et al., 1999 [125]
1998	Brazil	Tabocas Reservoir	O	*Raphidiopsis* (*Cylindrospermopsis*) *raciborskii*	MBA, PCOX, LC-MS/MS	STX, GTX6, dcSTX, NEO, dcNEO	Lab culture	Molica et al., 2002 [61]
1998–2000	New Zealand	Waikanae and Mataura Rivers	B/O	*“Oscillatoria-like”*	MBA, Neuroblastoma assay	STX, NEO		Hamill, 2001 [126]
2000	Brazil	Armando Ribeiro Gonçalves Reservoir and Pataxó channel	O	*-*	preCOX	C1,2, GTX (?), GTX5 (B1), STX		Costa et al., 2006 [127]
2000	Portugal	Lake Crato	O	*Aphanizomenon gracile*	PCOX	STX, NEO, GTX1	Natural and Cultured	Pereira et al., 2004 [128]
2000	Brazil	Billings Reservoir, Sao Paulo	O	*Microcystis aeruginosa*	MBA, PCOX, LC-MS/MS	GTX1,2,3,4	Lab culture	Sant’Anna et al., 2011 [107]
2002	Brazil	Tapacurá Reservoir	O	*Dolichospermum* (*Anabaena*) *spiroides*	MBA, PCOX	STX, NEO, dcSTX	Natural and Cultured	Molica et al., 2005 [129]
2002–2003	New Zealand	Waikato River	O	-	ELISA (manufacturer unspecified)	-		Kouzminov et al., 2007 [130]
2003	China	Lake Dianchi	O	*Aphanizomenon* sp.	MBA, PCOX	STX, NEO, dcSTX, dcGTX2,3, GTX4	Natural and Cultured	Liu et al., 2006 [131]
2004	Brazil	Billings Reservoir, Sao Paulo	O	-	PCOX	STX, NEO, GTX2,3		dos Anjos et al., 2006 [132]
2005–2008	France	Champs-sur-Marne, Paris	O	*Aphanizomenon gracile*	Neuro-2a cell-based assay, LC-MS/MS	STX, NEO	Natural and Cultured	Ledreux et al., 2010 [133]
2006	Brazil	Lake Lagoa do Peri	O	-	PCOX	NEO, GTX4	Adsorption testing	Romero et al., 2014 [134]
2006	Bulgaria	Borovitsa Reservoir	O	-	HPLC, Ridascreen™ ELISA	STX		Teneva et al., 2010[135]
2006–2013	Canada	St. Lawrence River	B	*Microseira* (*Lyngbya*) *wollei*	LC-MS/MS	LWTX1	Only looked for LWTX1	Hudon et al., 2016 [13]
2008	Germany	5 lakes in NE Germany	O	*Aphanizomenon gracile*	Abraxis ELISA, LC-MS/MS	STX, NEO, GTX5, dcSTX	Lab culture	Ballot et al., 2010 [136]
2008 ^†^	Mexico	Lake Catemaco	O	-	Abraxis ELISA	-	Seston, water, and snails	Berry & Lind, 2010 [120]
2008–2009	Greece	Lake Pamvotis	O	-	Abraxis ELISA	-		Gkelis et al., 2014 [137]
2009	Artic	Northern Baffin Island near Cape Hatt	?	-	Abraxis ELISA, preCOX	-		Kleinteich et al., 2013 [138]
2009	Guatemala	Lake Atitlan	O	-	Abraxis ELISA	-		Rejmánková et al., 2011 [139]
2009–2010	Florida, United States	Silver Glen and Blue Hole Springs	B	*Microseira* (*Lyngbya*) *wollei*	LC-MS/MS	dcSTX, dcGTX2,3 LWTX1,2,3,4,5,6		Foss et al., 2012 [14]
2009–2011	Brazil	4 Reservoirs of Rio Grande do Norte	O	-	Beacon ELISA	-		Fonseca et al., 2015 [140]
2010	Russia	Lake Baikal	O	-	Abraxis ELISA	-		Belykh et al., 2015 [141]
2010	Russia	Lake Baikal and Reservoirs of the Angara River	O	-	Abraxis ELISA, MALDI-TOF	STX, NEO, dcGTX2/3, dcGTX1/4	Paired with above study	Belykh et al., 2015 [59]
2010	Australia	Murray and Edward River systems	O	-	preCOX, Abraxis ELISA, Jellet rapid test strips	-	<LOD except for 3 samples by ELISA	Bowling et al., 2013 [142]
2010	Canada	St. Lawrence River (Lake St. Louis)	B	*Microseira* (*Lyngbya*) *wollei*	LC-MS/MS, LC-QToF	LWTX1,6		Lajeunesse et al., 2012 [96]
2011	New Zealand	Drinking-water Reservoir and Groynes Lakes in South Island	B/O	*Heteroscytonema* cf. *crispum*	preCOX, Jellett PSP Rapid Test Kit	STX	Natural and Cultured	Smith et al., 2011 [143]
2011	Brazil	Mundau River basin and Araripe, Ceara	B	*Geitlerinema amphibium, Geitlerinema lemmermannii, Cylindrospermum stagnale and Phormidium uncinatum*	PCOX	NEO, STX, dcSTX, GTX1,4	Lab culture, suspect LC	Borges et al., 2015 [144]
2011	Brazil	Itupararanga Reservoir, São Paulo	O	*Raphidiopsis* (*Cylindrospermopsis*) *raciborskii*	Beacon ELISA	-	Reported STX quota per trichome and per L	Casali et al., 2017 [145]
2013–2015	Russia	Central European Russia and West Siberia	O	-	LC-MS/MS	STX		Chernova et al., 2017 [146]
2013–2015	Greece	Lake Karla and Kalamaki Reservoir	O	-	Abraxis ELISA, LC-MS/MS	STX, NEO		Gkelis et al., 2017 [147]
2014	Greece	Lake Vistonis	O	-	LC-MS/MS	STX, NEO		Moustaka-Gouni et al., 2017 [148]
2014–2017	New York, United States	Chautauqua Lake	O	-	PCOX, RBA, Abraxis ELISA	*	Unknown PSPTs	Smith, Z.J. et al., 2020 [149]
2015	Russia	Lake Baikal	B	-	Abraxis ELISA, MALDI-ToF	STX, NEO, GTX5, dcSTX, dcNEO, dcGTX2/3, dcGTX1/4, doGTX 2/3/4(?)	Constitutional isomers of doGTX2/3 not determined	Belykh et al., 2016 [59]
2016	Greece	Karla Reservoir	O	-	Abraxis ELISA	-	Pelican death	Papadimitriou et al., 2018 [150]
2016–2017	Brazil	Paraíba River Basin	O	-	Abraxis ELISA	-		dos Santos Silva et al., 2019 [151]
2017	New York, United States	Butterfield Lake	B	*Microseira* (*Lyngbya*) *wollei*	PCOX, Abraxis ELISA, LC-MS/MS	STX, GTX3,5(?), dcGTX2,3, dcSTX, LWTX2/3, LWTX5, *		Smith, Z.J. et al., 2019 [12]
2017	Canada	St. Lawrence River	B	*Microseira* (*Lyngbya*) *wollei*	LC-MS/MS	LWTX1	Only looked for LWTX-1	Poirier-Larabie et al., 2020 [152]
2017	Russia	Irkutsk Reservoir near Lake Baikal	O	-	LC-MS (TOF), Abraxis ELISA	STX	Precolumn modification of STX	Grachev et al., 2018 [153]
2017–2018	New York, United States	Cayuga Lake	O	-	PCOX, ELISA, RBA, LCMS	STX, *	Unknown PSPTs	Smith and Boyer, 2024 [60]
2018	Denmark	Northern Zealand Lakes	O	-	PCOX	STX, NEO, dcNEO, dcSTX, GTX2,3		Podduturi et al., 2021 [154]
2018–2019	Ohio, United States	Western Lake Erie	B/O	-	Abraxis ELISA	-		Nauman et al., 2024 [155]
2018–2019	South Carolina, United States	Lake Wateree	B	*Microseira wollei*	LC-MS/MS	LWTX1,4,5,6		Putnam et al., 2022 [156]
2019	South Carolina, United States	Lake Wateree	B/O	*Microseira wollei*	LC-MS/MS	LWTX1,4,5,6, dcSTX	Export, stability, and degradation study	Metz et al., 2022 [97]
2019	China	West Lake in Hangzhou	B	*Microseira wollei*	LC-MS/MS	STX, NEO, dcSTX, GTX2,3,5, dcGTX2,3, C1, LWTX1	Lab culture	Chen et al., 2024 [99]
2019	Multiple	Multiple	O		LC-MS/MS	35 analogs	Re-evaluate PSPT producers (in culture), PSPT structures incorrect	D’Agostino et al., 2019 [26]
2021	Kansas, United States	Blue River, Mill Creek, and Indian Creek	B	-	Abraxis ELISA	-	ELISA does not detect all toxins	Rider et al., 2024 [157]

Abbreviations: MBA = mouse bioassay; ELISA = enzyme-linked immunosorbent assay; preCOX = pre-column oxidation with liquid chromatography and fluorescence detection; PCOX = liquid chromatography (HPLC) post-column oxidation with fluorescence detection; RBA = receptor binding assay; LCMS = liquid chromatography with mass spectrometry (may include various LC and MS types); LC-MS/MS = liquid chromatography with a QQQ/triple-quadrupole mass detector; MALDI-ToF = matrix-assisted laser desorption ionization time of flight; LC-QToF = liquid chromatography quadrupole time of flight; FAB-MS = fast-atom bombardment mass spectrometry; NMR = nuclear magnetic resonance. ^†^ Collection date estimated from publication date. ( ) Reported/former taxa identification. / Mixture (Commas represent individual congeners while a / represents a mixture of isomers that were not or that could not be differentiated). ? Paper text was unclear. * Unknown PSPTs.

### 3.3. Historical Large-Scale Measurement of PSPTs

Freshwater PSPT monitoring and surveillance continues to grow in an effort to understand the impact of this group of toxins on public and economic health, water access and recreation, and food and potable water supplies at the regional, state, and national levels [158,159]. While limited, there are several studies that have implemented regional-sized or larger monitoring efforts for PSPTs (Table 3). Environmental PSPT concentrations have generally been low (<1 ppb STX eq.), with concentrations above this usually identified at a limited number of locations.

Comparing PSPT concentrations between studies is complicated by the diversity in reporting units used for PSPTs (see Section 7.5). This is an ongoing issue with marine PSPTs, where erroneous reporting units can have a significant impact on regulatory thresholds and toxicity equivalency factors (TEFs) [160]. Comparing between different freshwater PSPT studies becomes difficult when concentrations are reported as STX equivalents (eq.) after TEF adjustment, STX eq. as activity of total PSPTs compared to STX in a sample, or as PSPT concentrations per congener or summed for the total concentration of PSPTs. Inconsistencies in reporting units will cause problems trying to assess the risk to public health, and therefore, we are in full agreement with Turnbull et al., 2020 in establishing consistent reporting units for freshwater PSPTs (see Section 7.5) [160].

Approximately half of the monitoring for PSPTs primarily used ELISA as a measurement tool (which was usually Gold Standard Diagnostics/Abraxis), with many using at least one additional method for confirmation. ELISA can often report false negatives as seen in an analysis of a subset of samples in New York [4], where many samples were positive by alternative PSPT methods but were either non-detect or detected at much lower concentrations with the ELISA. While the use of additional analyses is an appropriate strategy to address the limitations of ELISA in PSPT quantification, the approach does not mitigate the likelihood of receiving a false negative from an STX-ELISA due to poor cross reactivity.

**Table 3 marinedrugs-23-00271-t003:** Summary of the literature with large scale surveys (>6 unique freshwater sources) PSPTs in lakes/reservoirs. Each ELISA specifies the manufacturer used—if no manufacturer was identified then only ELISA was listed in the manuscript.

State/Country	Lakes with PSPTs/Lakes Surveyed (%)	PSPT Range (Median) [µg PSPTs/L] *	Analytical Method	Notes
Australia [112]	12/12 (100%)	14.7–568.6	MBA, PCOX	
Brazil [161]	37/37 (100%)	0.1–0.63	Abraxis ELISA	Toxicology assessment by ELISA only
Bulgaria [162]	4/120 (3%)	0.01–2.5	HPLC-DAD, LCMS, ELISA, in vitro cytotests	Compiled data from 35 studies
Czech Republic [163]	2/19 (10.5%)	0.03–0.04 (0.03)	Abraxis ELISA	
Denmark [164]	8–11/96	6–224 (34.1) µg STX eq./g DW^−1^	MBA, PCOX	Net tows and seston study. Mostly STX and GTX4, others present
Finland [165]	50	13–1070	Jellett rapid PSP test strips, RBA (Na-Ch and SXPN), PCOX, LCMS	
Finland [166]	21/32 (66%) (13–59% of samples)	<0.01–1.47 (0.031)	PCOX	Mostly STX, some dcSTX
France [167]	10 (14% of samples)	<0.05	LC-MS/MS	Lakes with toxin not reported
Germany [168]	10/29 (34%)	NA	ELISA, LC-MS/MS	
Greece [137]	3/6 (50%)	0.4–1.2	Abraxis ELISA	
Greece [169]	5/14 (36%)	0.1–6.65 (0.73)	LC-MS/MS (various extraction/separation methods)	Median is sum of STX + NEO in entire sample set
New Zealand [170]	38/42 (90%)	0.001–0.99 (0.00113)	ELISA, Neuroblastoma Assay	Only analyzed toxigenic cyanobacteria
Ohio, United States [171]	25/105 (24%)	0.022–0.880	ELISA, LC-MS/MS	Includes water treatment finished water
Poland [172]	12/34 (35%)	<0.01–0.57	PCOX	Dead fish and shellfish also analyzed. Detected STX and dcSTX
Sweden [173]	98 sites/blooms (47% of samples)	72 ± 190.4 ^†^	LC-MS/MS	Unclear description of abundance and sample percentages. Only measured STX, NEO, dcNEO
United States [174]	6/1161 (<1%)	>0.2 (0.03)	Abraxis ELISA	
United States [175]	4/23 (17%)	0.02–0.2 (0.03)	ELISA	
New York, United States [4]	29–36/245	0.38–923 (6.1–14)	PCOX, ELISA, RBA, LC-MS/MS	Many unknown PSPT-like compounds
United States [176]	6/11 (55%)	ND-0.913	ELISA (and qPCR)	
Texas, United States [177]	8/20 (40%)	ND-0.016 (0.003)	LC-MS/MS	Only measured STX
Uruguay [178]	18	ND-14.62 (1.74)	Abraxis ELISA	

* Concentrations are reported in multiple ways, including PSPTs as a sum of congeners, STX eq. as a sum of toxicity after conversion by TEF, or STX eq. as a sum of PSPTs based on activity relative to STX. See Section 7.5 for discussion on unification of reporting units. ^†^ Only reported an average and standard deviation.

## 4. Overview of Detection Methods

Analytical detection of PSPTs is considerably more difficult than many other cyanobacterial toxins (e.g., microcystins, anatoxins, cylindrospermopsins, etc.). This is due to the inherent high polarity of PSPTs resulting from two charged guanidinium groups, as well as sulfate, sulfonate, and hydroxyl functional groups. Isomeric structures lead to further challenges as the chemical and biological behavior of PSPT stereoisomers is not identical. The earliest certified method for analysis of PSPTs was the mouse bioassay (AOAC method 959.08) from 1959 [179]. Although proven highly effectively and reliable, this method has garnered considerable protest from animal rights groups over the years and has fallen out of favor due to cost, ethical issues, and challenges in interpretation of toxicity [180]. Alternative techniques for the detection of PSPTs for human health and regulatory use include ELISA detection, HPLC coupled with fluorescence or mass spectrometric detection, and receptor binding assays. Each technique will be discussed individually in the ensuing sections.

While the mouse bioassay is still a commonly used regulatory method, it is not broadly applicable in freshwater systems due to the lower concentrations of toxin often found in these systems, as well as the expense and impracticality of the method for the bulk of important scientific questions and human health issues. It is therefore not described in detail. Similarly, while there are a number of sensors in development for PSPTs, such as surface plasmon resonance [181], these methods are not yet ready for use in a regulatory environment and usually have poor performance when looking across the suite of potential PSPT congeners, and therefore, they are not discussed. High-resolution mass spectrometers are being applied more frequently in non-target chemical environmental analysis but have not been widely used or validated for freshwater PSPTs, so discussion of mass spectrometry is mostly limited to quadrupole-based instrumentation. Genomic monitoring tools, such as those quantifying *sxtA* gene counts, are also in use in some localities. However, it is difficult to quantitatively establish protective guidelines relying on surrogates without an *a priori* investigation of the quantity, types of toxins, and toxicity of PSPTs in a freshwater system; therefore, they are not detailed here. As analytical data of PSPTs improve, genomic monitoring techniques will become more useful for human health risk assessment.

### 4.1. Receptor Binding Assay

The RBA is a competitive binding assay that detects the composite toxicity of all PSPTs in a sample based on binding affinities (and thus specific toxicity) to their biological receptor, thereby estimating a sample’s total toxic potency as a function of its PSPT mixture [182,183]. PSPT concentrations determined by this assay are expressed in STX binding equivalents, in which tritiated STX (^3^H-STX) competes with unlabeled STX, either reference standard or sample, to bind to a finite number of receptors. A few different STX binding proteins have been studied as the target receptor for this detection method, including voltage-gated sodium channels [184,185] and saxiphilin [186,187].

PSPT concentrations in extracts are determined with a sigmoidal dose–response curve, where total ligand binding of the radiolabeled STX is represented as β/β_max_ [184], or bound/maximum bound toxin. The resulting toxin–receptor complexes are then measured by the beta emissions of tritium using a liquid scintillation counter [188]. To achieve the highest accuracy of estimated STX equivalent concentration, samples are diluted to fall within the linear segment (0.2–0.7 β/β_max_) of the standard curve [184,188].

The receptor binding assay is an effective option for reporting an integrated toxicity value for a mixture of PSPTs using a single STX standard and therefore is an important tool for screening the toxic potential of seafood and water samples in the context of assessing human health risk [189]. The receptor binding assay has received Association of Official Analytical Chemists (AOAC) approval as an Official Method of Analysis (AOAC method 2011.27) for the detection of PSPTs in marine shellfish [184,188], but is not yet an official method for other matrices, including freshwater samples. Since the RBA is a functional assay that approximates PSPTs based on their binding affinities to a target receptor, its response varies based on the toxic potencies and concentrations of individual PSPT congeners present in a sample. As such, we would expect more toxic congeners, like STX and/or NEO, to have a greater response within the RBA and less toxic congeners, like LWTXs [190], to have a lesser response. The RBA should not detect low-toxicity congeners unless they are at sufficiently high concentrations, including congeners that can be converted to more toxic derivatives (e.g., C1–C4) [183]. However, false negatives may occur due to low binding affinity (and toxicity) of some PSPT congeners [183,191].

### 4.2. Saxitoxin and Neosaxitoxin Enzyme Linked Immunosorbent Assay (ELISA)

Key to the preparation of a successful immunoassay for STX and other PSPTs is the development of antibodies against the PSPT ring system. PSPTs are too small to elicit an immune response directly; therefore, the haptens must be conjugated to an appropriate carrier molecule to generate anti-toxin antibodies. These carrier molecules (generally proteins) can be coupled to PSPTs using a variety of approaches, including formaldehyde condensation by the Mannich reaction, reductive alkylation with periodic acid, or the glutaraldehyde reaction [192]. The anti-toxin antibodies are then produced using a toxin-immunogen conjugate (e.g., glucose oxidase [192] or keyhole limpet hemocyanin [193] coupled to STX or NEO) injected into mice [194] or rabbits [109,195]. Polyclonal antibodies can be purified using affinity chromatography [196,197], gel electrophoresis [195], or used directly for analysis.

STX is highly decorated around the core ring system in the different PSPT congeners (Figure 1). The cross-reactivity of antibodies to the individual PSPT variants depends on the enzyme-STX conjugate used to make those antibodies [108,198,199]. Different linkages of STX to the immunogen change the epitope, or binding site, on the PSPT antigen available for binding to antibodies, leading to highly variable cross-reactivity between congeners. Antibodies prepared using STX as the hapten have historically had poor cross reactivity with the NEO-based derivatives (e.g., NEO, GTX1, and GTX4) that are hydroxylated at the N-1 position (Figure 1) [108,200]. Due to the poor cross-reactivity, the combination of STX and NEO enzyme–toxin conjugates and antibodies were trialed to increase ELISA accuracy but have not been put into use in currently commercially available kits [198,201], and modern PSPT ELISAs still struggle with poor cross-reactivity [202]. PSPT ELISAs have been developed in indirect [108,195,197,203] or competitive [197,198,203] formats, although the most commonly used and commercially available are competitive ELISAs, using the binding of free toxin versus a toxin–enzyme conjugate with STX or NEO [192] or GTXs [204] to an anti-toxin antibody. The Gold Standard Diagnostics/Abraxis STX-ELISA allows free toxins to compete with a STX-horseradish peroxidase (STX-HRP) conjugate for a rabbit anti-toxin antibody [109]. These STX-HRP rabbit antibody complexes are captured by an anti-rabbit antibody fixed to the ELISA plate. The addition of HRP substrate reacts with the captured HRP conjugate to give a product whose absorbance is quantified at 450 nm.

STX-ELISA is highly sensitive to structural modifications of STX [109]. The site(s) selected to link a PSPT to an immunogen for production of toxin antibodies and/or to enzyme conjugates is crucial as this step plays a major role in determining how reactive an ELISA will be to PSPTs other than STX. As congener patterns in freshwater PSPT producers can be highly variable between strains (Table 2), the STX-based ELISAs should be expected to have more variability compared to other PSPT analyses described here. Studies looking at correlations between PSPT concentrations measured by ELISA and other independent variables require extremely cautious interpretation because variability in toxin profile could lead to higher, lower, or no change in ELISA response, depending on the types and abundance of congeners present (see Section 7.1).

### 4.3. Oxidation and Fluorescence Detection with High Performance Liquid Chromatography

PSPTs with their polar zwitterionic nature are difficult to resolve using traditional reverse phase HPLC. The ring system lacks a suitable chromophore for detection by UV absorbance or fluorescence. For this reason, most early HPLC methods coupled ion-pair chromatography with post-column oxidation (PCOX) [205]. This chemical oxidation converted the PSPT backbone to a fluorescent derivative that could be detected using a grating fluorometer [206]. Early versions of the PCOX system required the use of three separate solvents with isocratic elution and therefore three injections for the resolution of all toxins for which reference standard materials were available [205]. This significantly reduced the efficiency of the method for high-throughput purposes (e.g., those needed in a regulatory environment). Newer iterations of PCOX employ solvent systems that can resolve most toxins in one analytical run, although the C-toxins must be resolved with a separate solvent system [206,207]. A second oxidation method uses pre-column oxidation (PreCOX) to convert PSPTs into fluorescent derivatives, followed by separation of the oxidation products by reverse phase HPLC. This allows for the separation and detection of PSPTs without the use of ion-pair reagents but produces a different set of challenges such as some compounds forming multiple products [208] and/or different PSPT congeners producing the same product [209].

Both the PreCOX and PCOX methods are AOAC-approved (AOAC methods 2005.06 and 2011.02, respectively) methods for the analysis of PSPTs in shellfish [207,209]. The reactivity of the PSPTs to oxidation is not congener specific, with oxidation able to react with many of the elucidated PSPT congeners to form fluorescent derivatives. This allows these techniques to detect toxins without individual standards, including the detection and quantification of the LWTX2,3,4,5,6 [12] or other unelucidated PSPTs with which there are no commercially available reference standards [29,60,61].

The fluorescent conversion of STX and related derivatives occurs through a Bayer–Villiger oxidation of the bond between the tetrahedral carbon at C-4 and the hydrated ketone at C-12, breaking open one ring of the PSPT ring system while the remaining two rings aromatize (Figure 2) [210,211]. Hydrolysis of the carbamoyl functional group from C-13 does occur but was not found to be necessary for the ring conversion [208]. Only a limited number of PSPTs have been investigated for the oxidation products that they produce, while the oxidation product ratios for different PSPTs have not been determined [208]. Variation in the concentration, pH, and type of the oxidant can change the oxidation reaction and analytical response of different PSPT congeners greatly [209,212,213].

Different functional groups heavily impact how well the reaction proceeds to produce fluorescent products. A sulfate at C-11 increases the response 3- to 5-fold over STX due to electron withdrawal from C-12, while hydroxylation at N-1 reduces the fluorescent response 2-fold (Table 4) [12]. STX derivatives containing a hydroxyl functional group at the C-12 position, rather than a hydrated ketone have a conversion efficiency lower than their diol counterparts, with LWTX1 and 12β-do-dcSTX converting to fluorescent derivatives 10 times less efficiently than STX when using periodic acid as an oxidant [12,27]. The oxidant used during early PCOX trials was hydrogen peroxide. It was originally proposed that the 12β-do-STX derivatives could not be converted to fluorescent derivatives [39]. However, replacement of hydrogen peroxide with periodic acid, currently the most commonly used oxidant for PCOX, allows for the detection of these compounds as periodic acid likely oxidizes the C-12 hydroxyl back to a ketone, which then reacts to form fluorescent products [214]. NEO and GTX1,4 PCOX response was significantly reduced relative to STX, likely due to it being more difficult to form an aromatized ring system [68]. N-1 hydroxylation impedes the formation of products **a** and **b** (Figure 2) because the -OH associated electrons are necessary for the rings to become aromatic. An additional N-1 reaction must occur, as oxidation products of hydroxylated PSPT derivatives do not differ from those that are not functionalized at N-1 [208]. Without losing the N-1 hydroxyl group, oxidation products would retain two adjacent and positively charged nitrogen atoms on the same ring system.

The presence of the ketone/diol at the C-12 position is a requirement for the PSPT ring system to be efficiently oxidized into a fluorescent derivative, which is an important consideration for the use of oxidation techniques when analyzing for LWTXs, M-toxins, and other related derivatives, which are reduced at this position. Changes in other functional groups increase or decrease the efficiency of the fluorescent conversion, where the C-11 sulfate on LWTX1 produced no increase in conversion relative to STX (9%) (Table 4), while 12β-do-dcSTX was also reported to have a response relative to STX of 10% [27]. This is notably different from GTX3, which is oxidized ~3.5–4.5× more efficiently than STX (Table 4). The efficiency of the fluorescent conversion was primarily limited by the oxidation of the C-12 hydroxyl back to a geminal diol/hydrated ketone, rather than the electron withdrawing from the 11-sulfate moiety.

### 4.4. Mass Spectrometry

Traditional reverse phase HPLC methods for PSPTs use ion pair reagents for retention due to the polar nature of PSPTs. These ion-pair reagents are incompatible with mass spectrometry as they cause ion suppression and source contamination. PSPTs can bind so strongly to normal phase stationary phases that they require very high salt concentrations for elution, which is also incompatible with mass spectrometric detection [43]. In recent decades, the development of columns using hydrophilic interaction liquid chromatography (HILIC) technology has made it possible to separate highly charged and polar compounds that are difficult to isolate in reverse phase systems [217].

Recent improvements in analytical methodology for the detection of PSPTs have revolved around improved detection by mass spectrometry utilizing advances in HILIC chemistry. Coupled with the new generation of mass spectrometers, this has made methods more sensitive, selective, and flexible. Analytical methods for PSPTs utilizing a range of HILIC-MS/MS column technology include: Dell’Aversano et al. [75], Boundy et al. [218], Diener et al. [219], Turrel et al. [220], Cho et al. [221], Ciminiello et al. [222], Thomas et al. [223], Blay et al. [224], Halme et al. [225], Bragg et al. [226], Lajeunesse et al. [96], Ballot et al. [136], Foss et al. [14], Armstrong et al. [227], Beach [228], Qiu [229], and Smith et al. [230]. Single and inter-laboratory validation of a HILIC-MS/MS method for the detection/quantitation of PSPTs in shellfish were recently achieved [231,232]. These internationally validated mass spectrometry methods for PSPTs have the ability to detect the full suite of common marine PSPTs in a single analysis, often with analysis times under 10 min. Exemplifying the power of the new instrumentation and the improved detection limits of the detectors, the toxin profiles for six PSPT producing cyanobacteria cultures were recently reevaluated identifying a number of PSPTs that had not previously been detected using older analytical techniques [26]. It should be noted that all three of the LC methods (PCOX, PreCOX, LC-MS) detailed are significantly more expensive than RBA and ELISA in both cost and required expertise, while mass-spectrometric-based detection requires significantly more capital investment than the two fluorescence-based chemical oxidation methods.

While mass spectrometric identification of PSPTs has shown tremendous promise for the identification of marine PSPTs, the availability of analytical standards has not kept up with the identification of new compounds (Figure 1 and Table 1). For the freshwater PSPTs, the discovery of the LWTXs exemplifies this issue, as the lack of commercially available standards limits the ability to accurately detect, identify, and quantify these compounds by mass spectrometry.

Modern high-resolution MS (HRMS) methodology may offer an alternative route to identifying PSPTs where reference materials are unavailable. To date, HRMS analyses of PSPTs have been demonstrated in shellfish [228,233,234,235] and human matrices [236]. These HRMS systems can be designed and validated for the detection of PSPTs; however, their capabilities for non-target analyses of PSPTs have not been well evaluated. Liang et al. designed a screening protocol for PSPTs, identifying 17 fragment ions that could be used to identify structurally similar compounds within the PSPT class [234]. HRMS methodologies have shown significant promise for detecting a wide variety of organic environmental pollutants, but the application of HRMS as a “non-target” PSPT detection method requires testing to validate its effectiveness and to determine how it works in comparison to RBA and/or chemical oxidation, which can also be used as “non-target” PSPT detection methods. A significant advantage of HRMS over these other PSPT methods is that it provides structural information that can aid in identification.

## 5. Selecting an Appropriate Method for Freshwater Paralytic Shellfish Poisoning Toxin Analyses

### 5.1. Requirements for Selecting an Analytical Method

Analysis of freshwater PSPTs can include challenges not observed in the analysis of PSPTs in shellfish and marine environments. These include: the wide variety of PSPT profiles observed in freshwater systems (Table 2), extraction of PSPTs from a wide variety of matrices, comparison and interpretation of results from different targeted and non-targeted methods for PSPTs, and the level of precision, accuracy, and congener profile specificity needed to address each set of scientific questions.

PSPT analysis is currently limited by a lack of standard availability. Certified reference materials are available for most of the common marine PSPTs, namely, C1,2, dcGTX2,3, dcNEO, dcSTX, GTX1-6, STX, NEO, and LWTX1 from the National Research Council (NRC), Canada [237], while others including C3/C4 are available from CIFGA Standards [238]. However, there are 60+ other PSPT congeners for which there are no standards available, even as uncalibrated qualification standards with or without impurities. While many of the congeners without standards are believed to be “rare” in marine systems or found in low abundance, there has only been limited exploration of freshwater PSPT profiles. The LWTXs are a significant concern in the United States, as during preparation of this review, the only commercially available reference material available was for LWTX1, whereas the other five LWTXs have been found in high abundance in strains of *Microseira wollei* from Florida [14], New York [12], South Carolina [230], and Alabama [15,121], and these cyanobacteria are likely present and producing PSPTs in other parts of the world.

Mass spectrometry, followed by ELISA, are the most common analytical tools for most cyanobacterial toxins (e.g., microcystins, anatoxins, etc.). However, unlike these other cyanobacterial toxins, the four PSPT methods described in Section 4 cannot be easily ranked against each other to determine the “best” method, as each has advantages and disadvantages relative to the others. What method will perform best and provide the most relevant data will depend on the scientific and/or public health questions of interest.

For the purpose of PSPT analysis, current methods utilizing MS-based detection can be thought of as “targeted” methods, where they perform best with PSPTs for which the analyst already has structural information about the compound and ideally a reference material for source optimization and fragment investigation. In these cases, modern MS methods for PSPTs generally provide better sensitivity, specificity, and often speed for PSPT analysis relative to other methods. The other methods: RBA, ELISA, and chemical oxidation, do not target individual toxins (through compound-specific molecular weight) but instead utilize non-specific properties related to the PSPT-ring system. These methods can be viewed as “non-targeted” because they can detect PSPTs for which standards are not available or that have uncertain molecular structure.

Concentrations of PSPTs in freshwater tend to be low (ppb) and require advanced techniques to reach the required detection limit. While the total concentration of individual PSPTs may be low, the summed amount of many low-concentration toxin congeners may exceed lower drinking water safety limits. Method preparation and instrument optimization both play an important role in achieving the necessary sensitivity to detect the low levels of PSPTs often reported in freshwater.

### 5.2. PSPT Method Limitations and Practical Application—Confirming Toxin Presence and Interpretating Results

An important set of considerations when viewing analytical PSPT results from a freshwater source are (1) has the methodology confirmed the presence of toxins with multiple lines of evidence and (2) has the method failed to detect PSPTs that are present in a sample? None of the analytical techniques for PSPT can detect all, and possibly even a large portion (>50%), of the known PSPTs with high sensitivity and high specificity. The questions being asked in each investigation dictates what method(s) will work best and, most importantly, what conclusions can be drawn from the analysis.

#### 5.2.1. Receptor Binding Assay

RBA targets PSPTs that bind to sodium channels and therefore provides the best sensitivity for toxins that bind strongly to this target, such as STX. Conversely, this method is less effective for low- or moderate-toxicity PSPT variants (e.g., some of the C-toxins), which do not strongly bind to sodium channels. This method may underestimate the true concentration of PSPTs present, as the majority of known PSPTs have worse sodium channel affinity relative to STX [239], the congener standardized against in the AOAC certified RBA method [184]. The RBA analytical methodology was not designed to report PSPT concentrations but to report “STX equivalents as a determination of toxic potency” (personal communication, Greg Doucette). Great care should be taken when interpreting RBA results in freshwater environments within the context of this statement. While this detection method is advantageous for using the functionality of PSPT congeners to determine the composite toxicity in a sample, it should not replace analytical techniques where chemical characterization is appropriate [183]. A non-detect result by RBA does not mean that PSPTs are absent, only that there was no detection of voltage-gated sodium channel inhibition above the LOD, but there may still be significant concentrations of PSPTs present.

There were very significant differences in the relative binding affinities of PSPTs measured by Usup et al. [191] and Zakrzewska et al. [216] using the RBA (Table 4). Notable differences were a ~3× increase for NEO, a 5× increase for dcSTX, and a 2× decrease for GTX1,4 with the reasons for the discrepancy uncertain. The difference between these two studies is a reflection on the reliability of PSPT analyses and why there need to be standardized performance metrics, field testing, and inter-laboratory validation studies even for previously certified methods.

A few studies have utilized the RBA as a preliminary and/or secondary method for determining total toxicity of environmental and cultured samples [121,149,165]. However, further investigation into the sensitivity of the RBA to freshwater matrices is needed to build upon the RBA work by Smith in 2019 [12], 2020 [4], and 2024 [60] and Llewellyn et al. in 2001 [187]. Performance of the RBA with freshwater matrices may be different to the performance in marine and shellfish matrices. Sample preparations with various collection techniques, physical cell disruption methods, and solvents should be explored to gauge matrix effects and determine the most suitable sampling and extraction protocols because the RBA has only been validated in marine environments. A well-characterized and validated protocol for freshwater PSPT analysis by RBA has potential for being a valuable tool for monitoring recreational and drinking waters, especially since freshwater PSPT chemical structures are largely unknown compared to marine PSPTs.

#### 5.2.2. STX ELISA

ELISA measures the binding affinity between PSPT congeners and an antibody tuned against a PSPT congener (most commonly STX). At the time of publication, commercially available ELISAs are designed to target STX, although other targets have been used. Commercially available ELISAs for PSPTs generally have had poor reactivity to congeners other than their primary target (e.g., STX, NEO), where every additional functional group change “away” from the original target reduces antibody binding significantly. For the Gold Standard Diagnostics/Abraxis ELISA, the highest cross-reactivity reported is gonyautoxin 2,3 at ~35%. Congeners with more than one functional group change, such as GTX1,4, reduces cross-reactivity to no more than ~10%. Because of this effect, the sensitivity of the assay is significantly hampered and may report non-detect even when PSPTs are present. Antibody binding affinity does not correlate with the toxicity (as the TEF of each PSPT congener), so concentrations measured by this assay may be higher or lower than the either the “true” or TEF-adjusted concentrations of a sample, and the only way to evaluate this effect is to measure the congener profile of the sample being evaluated. Qualitative assessment of magnitude is also hampered because changes in congener profile can mask or amplify an ELISA response, and ELISA by itself gives no structural information about PSPTs present (see Section 7.1).

#### 5.2.3. Chemical Oxidation (PCOX and PreCOX)

LC-based methods for PSPTs have a major advantage over assay-style detection methodologies in that they give congener-specific information about PSPTs in a sample by chromatographically separating components. While these methods are excellent for detecting toxins, they require expert level analysts for long-term operation and significant up-front capital investment, as well as additional maintenance costs.

The two oxidation-based detection methods have some advantages and disadvantages over each other, which have been detailed thoroughly for regulatory purposes [240,241,242]. However, both share similar drawbacks related to the oxidation efficiency of different PSPT congeners, where much of the uncertainty surrounding PSPT concentrations in freshwater environments is derived. PSPTs are converted to fluorescent derivatives (Figure 2) in oxidizing conditions, and this reaction is enhanced or diminished through functional group modifications on the PSPT molecule. Some functional group changes cause the molecule to poorly oxidize or favor reactions that lead to non-fluorescent oxidation products. As PCOX relies on a chemical reaction for performance, the sensitivity of the analytical method can vary based on the reaction conditions (concentration, pH, temperature, time, etc.) [243]. The manufacturer and/or source of the periodic acid was also found to impact reaction efficiency by reducing the conversion into fluorescent products even when using equivalent grades of periodic acid reagent (unpublished data). Poor oxidation leads to poor response in PCOX and PreCOX and, by extension, reduces both the sensitivity (increased LODs) of the analytical methodologies. A major loss in sensitivity could produce false negatives for some congeners, which can be important in some cases, such as investigating the metabolic pathways of cyanobacterial genera as described in D’Agostino et al., 2019 [26]. Sensitivity loss may also cause risk assessment problems when individual congeners are found in low abundance but sum together to exceed regulatory consumption limits, which are quite low (see Section 7.4).

In quantitative analyses, any variation in reaction efficiency can be normalized by using authentic reference materials as standards. However, since the majority of PSPT congeners do not have commercially available reference materials, quantitation estimates become more difficult due to the complexity of the reaction. This may produce inconsistencies in qualitative and quantitative analyses due to inconsistency in batch-to-batch preparation of oxidation and LC solvents.

Structural identification in PCOX and PreCOX is performed by a combination of fluorescent character and retention time. Retention time identification is precise for many PSPTs with reference materials due to the selectivity of the separations in PCOX and PreCOX but can become problematic when these standards are not available. In a sample of PSPT producing *Microseira* analyzed by the authors and containing LWTX5, doSTX, and 11α/β-OH dcSTX (dcM2α/β) (the specific isomer was not identified) as determined by LC-MS/MS, it was difficult to assign peaks to each compound in PCOX chromatograms based on chemical structure/polarity because of the similarities in each of these congeners [4]. Peak identification is hampered by the ion-pair reagent, which changes the separation chemistry significantly compared to “standard” reverse phase chromatographic separation systems. Furthermore, it is possible to misidentify PSPTs that co-elute or elute at very similar retention times in both oxidation methodologies (see end of Section 2.1 related to “GTX5”), while some co-eluting fluorescent peaks in PCOX and PreCOX may not be PSPTs [104].

It is possible and recommended to include a second level of structural identification and toxin confirmation with PCOX and PreCOX as (most) PSPTs only fluoresce after proceeding through an oxidation reaction followed by an acid modifier. When samples are run paired with and without an oxidant, the chromatograms can be compared to determine if there are peaks present in an oxidation chromatogram that are not present in one where water is run in place of the oxidant. This method can be used to eliminate fluorescent matrix interferences and determine whether there are PSPT-like compounds with elution times different than those of our limited range of standards, as the oxidation reaction can convert many of the known PSPTs into fluorescent derivatives. Relative retention time windows can be used to determine whether the compounds are GTX-like (e.g., sulfated) derivatives or STX-like (non-sulfated) in PCOX due to the major differences in elution strength of the A versus B solvents.

It must be noted that some PSPTs, notably the N-1 hydroxy congeners, appear to fluoresce more strongly in conditions without an oxidant in post-column chemical oxidation [104,105]. Based on known chemistry of the PSPT ring system, it is unclear why some PSPTs fluoresce without the addition of an oxidant. Fluorescence character is likely not present in pure standards of GTX1,4; it has been hypothesized that there are trace amounts of oxidant left behind in PCOX HPLC pump and tubing systems to produce this effect (Yuko Cho, personal communication), but it is unclear why the reaction chemistry would produce a higher response when the oxidant is present at trace quantities rather than the concentrations used during standard analyzing conditions. In our laboratory during PCOX analysis, when implementing a switch from the oxidant solution to water, the pump head and check valves had to be cleaned prior to paired “no-oxidation” analyses—oxidation of many of our PSPT standards could be observed to persist long after all of the oxidant had been flushed from the HPLC lines but without prior cleaning of the pump system. Not every group identified this sort of response, with Lim et al. reporting no GTX1,4 peaks in their chromatograms following the removal of oxidant from their system [54]. PreCOX has significant advantages over PCOX in this regard, where NEO did not show up as fluorescent without an oxidant in C-18 SPE cleaned mussel extract [243].

#### 5.2.4. Mass Spectrometry

Mass spectrometric methods for the identification of PSPTs are becoming more popular for routine PSPT measurement due to improvements in column technology and more powerful instrumentation. Mass spectrometry has a primary advantage over other methods due to determining molecular weight, an excellent confirmatory tool alongside retention time and peak shape for PSPTs. It is also more flexible in its ability to detect a wide range of PSPTs, as the system can be tuned for specific compounds, although there are limitations to this based on the number of compounds being measured, their elution, and the sensitivity requirements for the method. The largest impediment to mass spectrometry is that it is a “targeted” tool, where individual compounds are selected for by mass with the instrument optimized for their analysis. This may cause problems in freshwater systems as PSPT profiles are poorly described. Since PSPTs can have a range of optimal tuning conditions with complex chromatography, it can be difficult to screen for compounds without authentic standards.

Mass spectrometry currently relies on authentic standards for identification and confirmation, as there are limited ways to build secondary evidence for the structure of suspect PSPTs by the technique (e.g., universal fragment MS/MS fragments, diagnostic UV spectra). The fragmentation patterns for PSPTs can be complex, making it challenging to use spectra for identification with an unknown precursor mass [75], while the ionization mode and fragmentation efficiency of PSPTs can vary widely in an MS source inlet [218]. Method optimization is crucial, where similar mass spectrometric methodologies can produce a wide range in LOD for the same congeners (Appendix A).

### 5.3. Monitoring Recommendations for PSPTs

Not all scientific assessments of PSPT need to accurately determine toxic potential, but it is essential that monitoring studies of PSPT have some human risk assessment as a component of the program. Monitoring projects can become expensive when scaled and are forced to balance the rapidity and reliability of analysis against the expenses and the data requirements of the program. Freshwater cyanotoxin programs utilize ELISA for toxins like microcystins because it is affordable and relatively easy to run while providing adequate risk assessment [244,245]. This is not true for the PSPTs as described in previous sections and detailed in Section 7.1.

We recommend that any freshwater PSPT monitoring and/or analysis program include secondary methods in their protocols because it is nearly impossible to draw confident conclusions about risk when using only one of the analytical methods described. With the current state of PSPT analytical methodology, each method will leave gaps in information related to toxin presence/absence, congener distribution and concentration, toxic potential, and/or the “true” total concentration of PSPTs. We recommend incorporating RBA as one of the two methods for monitoring as it is far more affordable than the HPLC based methods and is on par with ELISA in difficulty and cost while providing direct measurements of exposure risk (Tod Leighfield, personal communication). In the United States, laboratories without a radioactivity license can order and work with small amounts of ^3^H-STX ordered through American Radiolabeled Chemicals, Inc. (ARC) (St. Louis, Missouri), as ARC has an exempt license from the Nuclear Regulatory Commission authorizing laboratories to order 50 µCi of STX tracer twice per year (ARC, Inc., personal communication). Each 96-well RBA plate requires approximately 2 µCi of ^3^H-STX (Tod Leighfield, personal communication).

Due to the number of PSPT congeners that have been structurally elucidated, limitations on quadrupole MS technology make it difficult or impractical to have a single method capable of detecting a large suite of analytes while retaining the required sensitivity. However, chemical oxidation methods can be an excellent tool to pair with MS detection for novel cyanobacterial or algal strains or for aquatic systems that have not been screened for PSPTs. PCOX and PreCOX are both highly sensitive and, even for unknowns, can identify the class of toxin with high precision (e.g., saxitoxin-like, sulfated gonyautoxin-like, C-toxin, etc.). Evaluating an unknown PSPT profile can be aided by using a combination of MS and fluorescence methods for screening, where both methods provide high sensitivity, specificity, and structural information. Components that might be missed with a targeted MS method can be first screened by chemical oxidation, with that information used to determine what components and/or congeners are best targeted by complementary MS-based measurement.

## 6. Toxicology and Human Health Concerns from Freshwater PSPTs

### 6.1. Acute and Chronic/Sub-Chronic Exposure to PSPTs

Acute consumption of PSPTs leads to PSP syndrome, which has been reported worldwide [1,246,247,248,249,250,251,252,253,254,255,256,257,258]. Meyer et al. in 1928 produced the first epidemiological description of PSP syndrome with symptoms including numbness of extremities, vomiting, diarrhea, floating sensation, ataxia, loss of coordination, and absence of deep reflexes [1]. Modern case reports of poisoning identify similar symptoms including but not limited to numbness, tingling, headache, vomiting, diarrhea, motor weakness, respiratory distress, and cardiac arrest [259].

Meyer’s work was the first compilation of PSP case reports, where they were investigating a series of poisonings in San Francisco, California, United States (US) over the previous year. This report described a systemic poisoning event in the US, as well as a synthesis of the case reports from around the world that seemed symptomatically linked to the San Francisco poisoning event [1]. Further epidemiological reports continued as subsequent blooms continued to appear in California, leading to sickness and death [260,261,262]. The work by Meyer and colleagues alerted health authorities to the potential disease, resulting in the increased incidence of PSP syndrome in the literature. While PSP poisoning was not well-known to science, traditional knowledge about the danger of consuming shellfish, due to some form of poisoning, was not new in some coastal communities [8,263].

The “gold standard” monitoring test for PSPTs in marine systems is the MBA, where every certified method for PSPTs references and/or is correlated against MBA to determine its accuracy and use in safety assessment. The MBA was developed using time-to-death measurements in mice outlined in a 1937 paper by Sommer and Meyer and described in detail in AOAC method 959.08 [179,264]. With their measurements, a relationship of dose and time-to-death between 5–7 min produced a relatively linear response. Concentrations of toxin by the MBA are determined by dosing mice with a concentration where death occurs between this part of the curve, and concentrations are reported as mouse units (MU). These early foundations still form the bedrock for environmental health testing of PSPTs for preventing acute intoxication.

The toxin is administered in the MBA through intraperitoneal injection (i.p.), which has been historically used to determine PSPT congener activity and/or toxicity [20,239,265]. However, i.p. injection does not always correlate to toxicity measured by the actual routes of human exposure, e.g., oral consumption (Table 5), and therefore, determination of PSPT TEFs through i.p alone is inaccurate [18,266]. Although this has been rectified with new toxicological data for many congeners [266,267], the majority of PSPTs have not been subjected to any thorough toxicological investigation, with many never having been tested by any method or assay. Whereas congener toxicity has been measured independently and each is converted into a sum toxicity in STX eq. using TEFs with human reference dose determined from STX’s toxicological profile.

The European Food Safety Authority (EFSA) panel established an acute reference dose for STX of 0.5 μg STX equivalents/kg b.w. based on an estimated no-observed-adverse-effect level (NOAEL) estimated from a lowest-observed-adverse-effect level (LOAEL) of ~1.5 μg STX equivalents/kg b.w. that had been measured in subset of poisonings in >500 people [215,239]. In the majority of the cases evaluated by the EFSA, concentrations of PSPTs in food consumed by individuals afflicted with PSP were determined by the MBA. There are conflicting views on the appropriateness of this number: Arnich and Thébault suggest through their models that 10% of cases exposed to 0.37 μg STX equivalents/kg b.w., e.g., a lowest-observed-effect level in contrast to the *higher* NOAEL by the EFSA, would have some sort of symptom [268]. Boente-Juncal et al. performed a feeding study with tetrodotoxin and STX, finding significant adverse effects (death) in mice at low (56.6 μg STX + tetrodotoxin/kg b.w.) doses [269]. In contrast, Finch et al. determined through a 28-day feeding study an NOAEL ~20× lower [270] than that suggested by Arnich and Thébault and no adverse effects on their mice at a dose 7× higher (715 μg STX.2HCl eq/kg b.w.) than Boente-Juncal et al. While the 80 µg STX eq./100 g shellfish food safety cutoff has been largely protective of the public, with no widespread poisonings from commercially tested and sold shellfish products, there is uncertainty into how this threshold was originally determined [271]. As the oldest regulatory threshold for PSPTs, it is believed that the threshold was determined through the work executed by Meyer, Sommer, and other colleagues on PSPTs through the 1920s–1940s. However, the certainty of whether this level is fully protective of all consumers is unclear as it is exceptionally difficult to acquire precise and accurate case reporting and toxin data from human PSPT poisoning events [272].

There is clear disagreement in the scientific literature about what is “safe” with regard to exposure to PSPTs (in terms of NOAEL) due to the complexity of extrapolating from a mouse model and the inherent difficulty of evaluating exposure and toxic potential in human poisoning cases. In the context of freshwater PSPTs, this becomes far more concerning because the routes of exposure (e.g., drinking water, swimming), concentrations and duration of exposure, and types of PSPT congeners may deviate from those found or assumed in marine systems where the bulk of toxicological information regarding PSPTs has been derived. There remain some uncertainties about what the true NOAEL for STX should be and whether there would be divergence depending on the endpoint(s) selected for the study.

Safety regulations for PSPTs have generally been successful at protecting consumers from poisonings from contaminated shellfish products, although there are still thousands of poisonings annually with a fatality rate in 1993 estimated to be 15% [7]. Between 1985 to 2018, 35% of all seafood toxin illnesses were reported to be from PSPTs (reports were submitted through Harmful Algae Event Database), with 3719 adverse events and 213 fatalities (5.7%) [258]. Doses that have produced oral intoxication are estimated to range from 144 to 1660 µg STX eq. per person, while lethal doses are estimated to range from 456–12,400 µg STX eq. per person [190]. The likelihood of exposure to a lethal dose in freshwater environments, such as during recreational water use or from consuming contaminated drinking water, is very low.

### 6.2. Medical Treatment of PSP Syndrome

To our knowledge, a summary of clinical treatment for PSP has not been detailed in the readily accessible literature or book chapters, and therefore, we report on a set of case reports and recommendations for treating patients suffering from acute exposure to PSPTs. Health impacts from non-life threatening and/or chronic exposure are poorly understood, and therefore, there are no established treatment options or protocols. Treatments for PSP syndrome mostly involve supportive care and management of poisoning symptoms as there are no direct antidotes to STX or its analogs. Patient monitoring and stabilization are the primary response in acute emergency room care, with the addition of intubation in severe poisoning events due to the impact of PSP syndrome on the respiratory system [273].

There is some disagreement on whether gastric lavage is recommended. Mines et al. recommended the use of lavage if a patient presents with symptoms within a few hours of ingestion [274], while Kao recommended the use of a gastric lavage or emesis if no vomiting had yet occurred [273]. Activated charcoal is recommended by both doctors as an effective treatment, although the mechanism may not be straightforward as PSPTs do not usually bind to non-polar substrate.

Dialysis may also be able to remove these low molecular weight, water soluble toxins, with marked improvement in a pair of patients reported by Acres and Gray [246], whereas Lan et al. reported dialysis in a renally impaired patient showing symptoms of TTX ingestion [275]. The hypothesized mechanism of TTX detoxification may also apply to PSPTs due to their chemically similar properties: low molecular weight, charged and highly water soluble, and lack of significant protein binding [275].

Lawrence et al. suggested a treatment for TTX poisoning, a toxin presenting clinical effects similar to PSP syndrome, by admitting all patients with, “Numbness of tongue, face, and other areas (distal); early motor paralysis and incoordination; slurred speech; normal reflexes” or worse symptoms [276]. Lawrence goes on to state that patients with severe symptoms, such as those with respiratory failure or paralysis, be mechanically intubated and atropine used to treat bradycardia.

### 6.3. Toxicological Gaps and Concerns

The acute dosing regimens studying PSPT symptomology are difficult to apply to the sub-acute concentrations of PSPTs more commonly identified in freshwater systems. The routes of exposure to freshwater consumers and users differ from those in marine systems, where poisoning occurs primarily through consumption of contaminated shellfish and crustaceans. Most toxicology and the derivation of reference doses has been tested through single-event doses rather than the medium to long-term doses expected through freshwater. While symptoms of acute poisoning are believed to resolve within a short time frame [277] and most nerve function appears to return within weeks [252,253], the effects from long-term exposure to PSPTs is unclear, as well as whether there are any interventions to minimize harmful impacts to bodily health following toxin consumption.

The MBA is the most common method of testing food for PSPTs, specifically when measuring absolute toxicity in products meant for human consumption. However, time-to-death curves for different PSPT congeners can look very different [266]. While we primarily use a small part (5–7 min) of this curve to calculate mouse units, the shape of the curve itself for different congeners can change in the range used for quantification by MBA, while outside of the range, the shape changes more drastically. This is further compounded by massive error bars, suggesting a highly variable physiological response between individuals.

Toxicological research is complex, and there are many challenges in ascertaining a reliable, robust, and accurate approach when designing animal model studies. However, identifying the correct dosing regimen is important for properly interpreting a human effect. In an acute exposure scenario, Finch et al. determined that the LD50 for STX was ~30% lower when given to mice on an empty stomach [278]. Although it makes sense that a dose in this scenario would not be very appropriate for a shellfish consumer, who are by definition not fasting, this becomes more complicated where freshwater users could be anywhere from fasted to full when consuming contaminated water. In the worst-case scenario, infants and toddlers who do not eat solid foods may have very different absorption characteristics than those found in adults, while children also develop glucuronidation through childhood, which may increase their sensitivity to PSPTs [279,280]. There is no toxicological information about the transport of PSPTs into breastmilk. Domoic acid, a low molecular weight and highly water-soluble toxin analogous to PSPTs, has been found to be transferred through breastmilk [281]. It is plausible that PSPTs could also be found in breastmilk as the toxins are known to distribute throughout the body [248], but it is unclear at what rate they would cross from plasma into breastmilk due to their high polarity and variable net charge [282].

A study dosing rats chronically with PSPTs at 3 and 9 µg/L (0.24 and 0.72 µg/day) identified multiple markers of oxidative stress [283], which may explain some of the neurological effects previously described during chronic dosing of PSPTs [284]. The low dose treatment consumed approximately the same quantity of toxin per kg of bodyweight as a 70 kg human consuming an 80 µg dose, the shellfish regulatory limit. Calculating a dose from drinking water depends on the quantity of water consumed per day, but at 2 L/day and ignoring the animal model comparison, this would suggest that oxidative stress could potentially be measured in a human if water concentrations were approximately 40 µg/L, well above most drinking and recreational water guidelines. This quantity of toxin is believed to be safe for an acute consumption scenario and is also unlikely to be a realistic long-term dose. While this dose does not exceed the estimated NOAEL for a 70 kg human (35 µg), it does exceed the 80 µg/100 g shellfish regulatory limits that are believed to be safe for short term exposure.

While the dose consumed by the rats was much higher than would be expected in drinking water, referencing this dose for a human exposure assessment would require, at minimum, one 10× safety multiplier due to species-to-species comparison, while another 10× could be reasonably incorporated for individual response sensitivity. As these types of symptoms have not been evaluated in human tissues and, in this case, required sacrifice of test subjects to measure neurological impact, not including additional safety factors would be an error. Incorporation of two safety factors would reduce the comparable human dose from 40 to 0.4 µg/L or lower, because effects were still measured in the rats in the low dose treatment. Based on the limitations of our understanding of non-acute cases of PSPT intoxication, there is strong support for incorporating safety factors on top of reassessment of the established NOAEL, which only includes an adjustment of 3× to calculate a NOAEL from a LOAEL and does not incorporate non-human toxicological data.

There is uncertainty in the total-body impact of PSPT intoxication because PSPTs can be found in so many organs [248] and bind to a wide range of voltage-gated sodium and calcium channels [280], and therefore, the endpoints chosen by a toxicology study can have a major impact on the results. When Finch et al. [270] performed their chronic dosing study with mice they looked at many endpoints—appearance and behavior, bodyweight, food intake, motor coordination and grip strength, cardiovasculature, blood chemistry and composition, and organ weight. In all of these cases, they saw no difference from the control and stated, “…the current PSP regulatory limit appears fit for purpose.” The study design and STX dosing by Ramos et al. [283] were similar but with a different set of endpoints and identified very distinct differences in oxidative stress markers. Because there are so many potential endpoints with PSPTs due to their broad activity to ion channels and systemic distribution in the body, it is critical to understand mechanisms of toxicity outside of those expressed in patients suffering from acute exposure or death, as this will have a major impact on determining an accurate NOAEL.

## 7. Protecting Human Health from Freshwater Paralytic Shellfish Poisoning Toxins

### 7.1. Quantification of Toxic Potential

Cyanobacterial toxins have been identified in drinking water facilities and have at times remained in finished water, with microcystins the most well-documented of these contaminants [285,286]. Cyanotoxins including PSPTs are also a recreational exposure risk from planktonic and benthic cyanobacterial producers, with the latter having been found to release toxins during shoreline drying [97]. Risk quantification of freshwater PSPTs is complicated by the variability of congeners reported in the literature derived from different detection methods, limited reference materials, and the natural variability between cyanobacterial strains (Table 2 and Table 3). Currently, concentrations of “old” (i.e., previously known) congeners represent the bulk of the toxins quantified, but cyanobacterial production of “new” emerging congeners of concern could be a significant percentage of total toxin [26]. For example, STX has historically been the primary target for monitoring as it is a common marine PSPT, but STX has not been demonstrated to be a major component of PSPT profiles in freshwater systems. The methods targeted at STX (or any one specific PSPT) may not reflect the integrated risk from the suite PSPT congeners produced by cyanobacteria. It is essential that water managers and scientists be able to quantify the concentrations of toxins and associated risks specific to contaminated drinking or recreational waters. Importantly, water quality management should not rely on the marine regulatory framework, which is protective of shellfish consumers [287] but may not be protective of human health in freshwater exposure scenarios.

The raw results from most PSPT methods are not generally directly applicable to human health and/or risk assessment. Differences in assay cross-reactivity, methodology and congener-specific toxicity all influence how a result is interpreted. The results from RBA analysis can be most directly extrapolated to risk, as concentrations reported by the assay can be directly correlated to MBA toxicity [182,183]. These numbers can be directly compared to STX reference doses and, by extension, consumption guidelines or limits established for human health and safety. Two studies measured relative binding affinity for PSPT congeners and reported very different affinities for PSPT congeners (Table 4) [191,216]. The source(s) of these differences must be determined since understanding the relative affinity for different PSPTs in this assay is important when determining potential toxicity for compounds without reference materials or TEFs. Variability in assay protocol that drives changes in relative affinity could put consumers at significant risk if the assay underestimates toxic potency. That said, the RBA has been repeatedly shown to protect human health and correlates well with the MBA even where there was significant diversity in PSPT profiles [287]. It is critical to note that the toxicity reported by the RBA assay is not the same as an in vivo assessment of toxicity (e.g., MBA, i.p. injection, oral dosing). Rather, the RBA correlates with the MBA directly without TEF conversions, modifications of the assay, or method output.

Quantification in PCOX and PreCOX is performed by comparison against authentic reference materials, where each congener can be converted into STX eq. using EFSA or FAO/WHO TEFs when a known compound is identified [215,239]. However, this becomes more challenging when a potential PSPT is identified without a calibration standard because method response is dependent on oxidation efficiency. Where a risk assessment is desired, quantification should be performed using the most comparable structural analog, noting that there are small differences in oxidation efficiency between isomers, and assume a TEF of 1. However, it must be recognized that this is a poor estimate as the true concentrations of toxin and TEFs could vary significantly (Table 5).

Toxicity estimates using mass spectrometry are performed similarly to that of chemical oxidation, where compounds that match authentic standards can be converted to STX eq. using TEFs. Mass spectrometric PSPT methods can function, albeit with significant difficulty, even when authentic PSPT standards are not available. Fragmentation and high resolution mass data can allow for the identification of compounds with a known molecular weight and structure with reasonably high confidence, but the ionization efficiency of PSPTs in electrospray ionization sources varies widely for some congeners even with small changes in cone voltage or collision energy [75,218]. The variance is hard to predict as the three-dimensional structure of PSPTs plays a role in the ionization efficiency and fragmentation of PSPTs, making it difficult to estimate concentration, even when using PSPTs that appear to be structurally similar when portrayed in two-dimensional space [223]. Quantification estimates should be performed similarly to the two oxidation methods, using the most analogous PSPT congener to estimate concentration and assuming a TEF of 1.

Current and commercially available ELISA kits should not be used to estimate human health risk for freshwater PSPTs without independent methodological verification. Several studies have compared different PSPT methods and found ELISA reported significantly different concentrations than other methods. One study with a subset of three samples from Australia measured by LC-MS/MS had concentrations 11–29× higher than the concentrations measured by ELISA, while one had a similar concentration and only GTX5 was detected [244]. A second study by Smith determined that ELISA concentrations were often several orders of magnitude lower or non-detect compared to PCOX and RBA [230], with the exception of samples from Cayuga Lake, NY, where a unique collection of unidentified PSPTs resulted in ELISA concentrations that were several times higher than PCOX and RBA [60]. Lastly, in a study with samples from three Ohio lakes, ELISA detected concentrations similar to those detected by LC-MS/MS in five samples, was lower by approximately half in four samples, while it significantly disagreed with RBA by several orders of magnitude in four of six samples (unpublished data). PSPTs with a lower binding affinity to ELISA antibodies may not have corresponding decreases in binding affinity to sodium channels and by extension could retain significant toxicity. Without profile information from another method, it becomes impossible to evaluate the “trueness” of a concentration measurement through an ELISA, even within orders of magnitude, because of the variance in cross-reactivity for different PSPT congeners (Table 6) [109].

Table 6 demonstrates how congener profile change can have an outsized impact on the ELISA concentration error relative to the true TEF-adjusted toxicity. There are also significant differences between the true sum of each PSPT congener and the ELISA result. Using Sample A as an initial reference, Samples B and C represent a shift in congener profile while the total amount of toxin remains the same, samples D and E represent a large shift in concentration for only one congener, and samples F and G represent a low and high quantity of a new congener into the PSPT profile. The errors calculated here would not be the same if other PSPTs were considered, as these errors vary depending on each congener’s relative reactivity and TEF.

### 7.2. Toxicity Equivalency Factors

Raw concentrations of PSPTs from most methods are not directly related to acute exposure risk but must be converted to STX eq. using prescribed TEFs [215,239]. These TEFs are most appropriate for acute exposure and, as discussed in Section 6.3, have uncertain relevance in chronic or sub-chronic scenarios. PSPT dosing studies at sub-chronic levels are limited and have been restricted to STX and TTX [269,270,283,288], and applications of these findings for the highest risk constituents, such as infants and children, are non-trivial.

While TEF conversions are paramount to achieving an effective risk assessment, there are some weaknesses in applying the TEF framework to a freshwater exposure model. PSPT congeners have different specific activities toward different types of sodium channels [215]. As PSPTs widely distribute throughout the body and bind to a variety of ion channels, other physiological changes may occur that have not been recognized as a symptom of acute poisoning (e.g., numbness and tingling, breathing issues, death, etc.). The TEFs for PSPTs are primarily derived from experiments related mostly closely to an acute exposure scenario, but a change in dose does not always appear to have a proportional physiological impact for the different congeners [266]. Similarly, PSPT congeners may have a greater or lesser effect on some organs with divergence from the existing TEFs that depends on tissue type. Estimating risk therefore could require assessing additional endpoints with chronic and sub-chronic exposure, where NOAELs and TEFs could be congener and organ dependent.

There are currently two sets of TEFs that can be used for converting individual PSPT congeners into an integrated toxicity. The first are those published by the EFSA in 2009 [239], and the second are from the FAO/WHO in 2016 [215], containing additional congeners and updates to some TEF values. While the newest set of TEFs is likely more accurate, the TEFs actually used in regulatory monitoring, such as those used by the US Interstate Shellfish Sanitation Conference [289] and the European Union [290,291], are the EFSA TEFs. The EFSA TEFs have been specifically incorporated into laws governing shellfish toxins and therefore have remained in use for food safety measurement even when the FAO/WHO TEFs have included more recent data into their assessment.

We suggest any newly established regulatory guidelines should use the more recent TEF standards in their assessments. We believe using the best available science in establishing regulatory guidelines will best protect human health, even if this diverges from the framework and existing standards set for marine food safety. There were important updates to TEFs in the 2016 FAO/WHO evaluation, such as the revision of the NEO TEF from 1 up to 2 based on the oral toxicity to mice, with relative potency by gavage, feeding, and i.p. injection being 1.7, 2.54, and 1.16, respectively [215,266]. Other PSPTs considered or modified were GTX5,6, dcSTX, dcNEO, and C1,3. While there is an existing framework in place for PSPT management in marine systems, this should not be duplicated for freshwater PSPT management as the use and availability of two sets of TEFs will only confuse stakeholders and make it harder to perform risk assessment.

### 7.3. Proposed Toxicity Equivalency Factors for M2α and M2β (“11-Hydroxysaxitoxin”) and dcM2α and dcM2β (“11-Hydroxydecarbamoylsaxitoxin”)

The FAO/WHO technical paper removed the “11-hydroxysaxitoxin” (M2) TEF because the compounds’ MBA activity, as reported by Schantz in 1986 [265], was actually for GTX2 and GTX3. How this mistake in the EFSA report happened is not entirely clear, but we believe it was likely a transcription error because the mouse units reported for 11α/β-OH STX in Table 13 of [239] are the same as those stated by Schantz in 1986 for GTX 2,3, with the error due to the similarity in names between 11-hydroxysaxitoxin and 11-hydroxysaxitoxin sulfate. However, in 1978, Boyer, Schantz, and Schnoes [20] purified or synthesized several other PSPT congeners, tested the toxicity [43] of the 11α/β-OH STX epimer pair and others, and reported these alongside that of GTX2 and GTX3. In the 1978 paper, they stated, “Hydrolysis of [GTX2/3], under conditions sufficiently mild…to leave the carbamate function intact (as determined by model experiments with [STX]) gave the equally toxic hydroxy-ketone [11α/β-OH STX]…Borohydride reduction of [11α/β-OH STX] or acid hydrolysis of [GTX2/3] led to the biologically inactive diol [11α/β(?),12-α/β(?)-hydroxysaxitoxin].” (It is not clear which or whether all of the four isomers were produced, separated, and tested). Appendix A, reproduced here and produced in 1980 by Boyer [43], reported the specific activity of 11α/β-OH STX to be 2300 MU/mg. Additionally, the potency relative to STX of 11α-OH STX to squid axon was found by Kao et al., 1985 to be 0.1 [292]. Another epimer pair not recorded in the EFSA TEFs was 11-hydroxydecarbamoylsaxitoxin (dcM2), which had an MU activity of 900 MU/mg.

These data suggest that the toxicity of the pair of 11-hydroxy STX isomers are similar to their GTX2,3 counterparts and should receive an equivalent TEF of 0.4 for 11α-hydroxy STX and 0.6 for 11β-hydroxy STX. For the 11-hydroxy dcSTX (dcM2) congeners, the TEFs would be 0.16 for 11α-hydroxy dcSTX and 0.23 for 11β-hydroxy dcSTX. Both of these TEF calculations make the assumption that if there was a mixture of both congeners in each bioassay, they would be in similar proportions to those found of a mixture of GTX2,3, which slowly epimerizes following purification [20,21]. We believe these TEFs should be added based on the consideration of concurrently accumulated data by the EFSA in 2009 and because this represents some of the only toxicity data available for these PSPTs. At the time of the 2009 EFSA report, the M-toxins had only first been identified in shellfish in the previous year [24], while the presence of the M-toxins as cyanobacterial and algal biosynthesis products would not be demonstrated for another decade. As these toxins are now known to be present in the environment, these potency data should be considered.

### 7.4. PSPT Regulatory Guidelines

Due to limited information on distributions of PSPTs in freshwater, regulatory limits for the PSPTs in drinking or recreational waters are not common. The few regulatory limits established for freshwater PSPTs are summarized in Table 7. While many of the drinking water regulations are similar, recreational thresholds in Ohio are lower than those found in the rest of the world as they established the lower limit of their calculation using bodyweight and drinking volume of children rather than an average adult. Decisions related to average size and bodyweight that establish a regulatory limit have large implications in determining actionable concentrations of toxin, as explored in Figure 3 and Box 1.

**Table 7 marinedrugs-23-00271-t007:** Representative drinking water and recreational exposure guidelines for PSPTs. All values are reported in µg/L.

Source	Drinking Water Exposure	Recreational (Short Term) Exposure	Recreational (Subchronic) Exposure
Australia [293]	3	-	-
Brazil [294]	3	-	-
New Zealand [295]	3	-	-
Oregon, United States [296,297] *	1 or 1.6 or 3	-	8 or 10
Ohio, United States [298,299]	0.3	3 (2016 only)	0.8
Utah, United States [300]	-	75	8
Washington, United States [301,302]	3	75	-

* Farrer et al., 2015 [296] established the formulae and tolerable daily intake used to calculate drinking and recreational water limits as 1 and 10 µg/L. The 2021 Oregon Public Health Authority document [297] cites a recreational use value of 8 and drinking water value of 1.6, while also stating the previous drinking water value was 3. This latter document does not cite the former, making it unclear which values Oregon public health had selected during this time period.

**Figure 3 marinedrugs-23-00271-f003:**
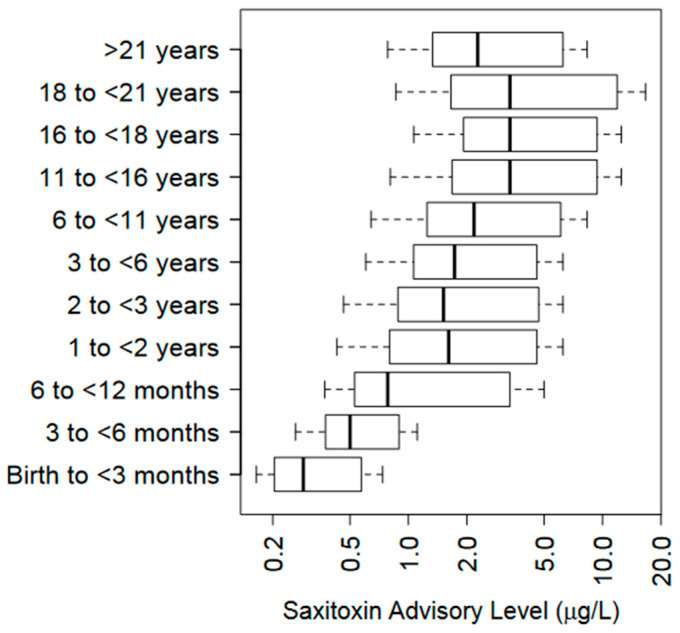
Variability in calculated saxitoxin advisory levels at all age levels and drinking water ingestion rates using a NOAEL of 0.5 µg/kg and an uncertainty factor of 10. Reproduced from Miller et al., 2017 [303].

Box 1Example of PSPT exposure level calculation for a bottle fed infant.PSPT exposure risk in drinking water may not be of
obvious concern due to the difference between consumption of high
concentrations in shellfish versus the likely much lower concentrations found
in drinking water. However, an example of a formula fed infant is
illuminating. Using the drinking water equivalent level (DWEL) set for
nitrate by the EPA* as a reference for drinking water formula for a 4 kg
infant drinking 0.64 L/day [304] and assuming the acute reference dose (ARfD) for shellfish is applicable (0.5 μg STX equivalents/kg b.w./day):

$DWEL=\frac{NOAEL\left(\frac{\mu g\;STX\;eq}{kg\;b.w./day}\right)\;infant\;b.w.}{L\;consumed}=\frac{\frac{0.5\;\mu g\;STX\;eq}{kg/day}\;4\;kg}{0.64\;L}=\frac{3.125\;\mu g\;STX\;eq}{L/day}$

This DWEL approximates the 0.3 μg STX eq./L DWEL established by the state of Ohio in 2016, United States after including a ×10 added safety factor due to data uncertainties. In 2020, Ohio revised their DWEL to 0.8 accommodating both children and adult TDI data.* the units in the EPA document are not the same as those used by the EFSA. The definitions and units written here are those described by the EFSA.

An example calculation is shown in Box 1 detailing the derivation of a drinking water threshold for a bottle-fed infant whose entire liquid diet is from a contaminated commercial source or utility. It should be noted that there is no toxicological information regarding transmission of PSPTs through breastmilk. Although the “safe” consumption rate calculation is approximately 0.3 μg STX eq./L for infants, this number changes significantly for larger and older individuals (Figure 3). Water intake rates are incorporated into these calculations as the water volume consumed per day accounts for weight. However, intake rates can increase significantly for individuals in some circumstances, such as exercising or heat stress, and in these cases, there may be a shift from chronic to sub-chronic or acute risk due to the high toxicity of PSPTs.

Protection limits should be established for all users, not only those consuming toxins in ideal scenarios. For example, national park visitors with high activity levels, such as hikers [305], may be more susceptible to health risks by toxin consumption as these users are more likely to intake large quantities of water while fasted in environments ripe for harmful planktonic algal and/or benthic cyanobacterial blooms, often without access to interventions that remove or detoxify PSPTs like those used in a public water source [306,307,308]. Furthermore, STX is approximately 30% more toxic to mice when dosed with the toxin while fasted [278], which is not an uncommon state for many people, for example, those who are exercising or who skip or delay breakfast. While regulations rightly make assumptions about the average size and quantity of a consumer to calculate tolerable daily intakes or other similar safety thresholds, this is not adequately protective of users that are the most at risk from consumption of PSPTs.

### 7.5. Reporting Units

Reporting units themselves are a frequent source of error, and sample toxicity is best expressed in molar (µM) units rather than by weight (µg) to minimize errors when “reporting or comparing toxicity” [160]. Summed results from ELISA should be reported as STX concentrations by ELISA, specifying the manufacturer of the ELISA kit, and avoid the term STX equivalents to not lead to confusion. Other PSPT methods, where possible, should report total PSPTs and STX eq. separately, after accounting for TEFs. Other sources of error that must be accounted for, and that are reviewed in detail by Turnbull et al., include “the units for calibration standards, raw concentrations and final results.…calculations should be explicitly described and the source of TEFs referenced”, and careful attention to presenting units of STX (free base) or STX.2HCl (or other counterions) for concentrations and in regulatory limits [160].

### 7.6. Exposure Risk Assessment from Consuming PSPTs in Freshwater Systems

A qualitative risk assessment with larger scale PSPT studies from Table 3 and the water quality guidelines from Table 7 suggests that there are exceedances of 0.3 µg PSPTs/L in freshwater systems worldwide. Occurrence percentages (and how they were reported) differed by study, but PSPTs were detected in a significant percentage (>20%) of waterbodies in 14/21 of the studies. Cyanobacterial blooms and related toxin concentrations are rarely normally distributed and are infamous for having extreme outliers with toxin concentrations orders of magnitude higher than regulatory thresholds [4]. To assess risk, median PSPT concentrations were reported, or when possible calculated, from the studies reported in Table 3 to better describe the typical concentrations observed. How common these exceedances occur is not entirely clear, but the majority of studies reported maximum concentrations of PSPTs above 0.3 µg PSPTs/L, and of the nine studies reporting median concentrations, four reported median concentrations of PSPTs well above 0.3 µg PSPTs/L. Finished water concentrations are presumably less than what is found in source water due to the use of flocculants, sedimentation, and filtration to remove intracellular/particulate PSPTs, while chlorination or other disinfection tools would serve to remove dissolved contaminants.

Three of four studies with median concentrations above 0.3 µg PSPTs/L used PCOX, while four of five studies with median concentrations below 0.3 µg PSPTs/L used ELISA. While limited, this suggests ELISA concentrations tend to report lower overall PSPT concentrations. Of the PCOX methods detecting regular (>50% of samples) exceedance of 0.3 µg PSPTs/L, this suggests that there may be chronic PSPT consumption concerns in some freshwater systems. Direct comparison of concentrations between studies with different detection methods is inadvisable, as the methods report PSPT concentrations differently (see Section 4 and Section 5).

While concentrations provide a cursory assessment of risk, this is unlikely to be accurate at determining toxicity because PSPT congeners require TEF adjustment prior to risk assessment. Of the 21 studies in Table 3, 11 reported substantial use of ELISA assays, which cannot be used to assess risk in freshwater PSPT systems. Unless STX was the dominant congener for studies utilizing ELISA as their primary assay, of which STX has not been demonstrated to be abundant in freshwater systems (Table 2), these assessments underestimated toxin concentrations, possibly quite significantly (Table 6).

Of the remaining studies not using ELISA as their primary method, three used LC-MS/MS and only measured: STX [177]; STX, dcSTX, and dcNEO [173]; or STX and NEO [169], and therefore, broad conclusions about risk should not be drawn. The fourth only used LC-MS/MS when ELISA concentrations exceeded drinking water thresholds, and therefore, if STX did not dominate the congener profile, this would likely have led to an under-utilization of LC-MS/MS to accurately determine toxin concentrations of known PSPTs [171].

The studies utilizing PCOX [4,112,164,166,172] used a common suite of STX and GTX standards, which can be used to calculate toxicity after TEF adjustment. However, one of these studies used seston net tows for collection, so retrospective extrapolation back to a water concentration is not possible with the data presented [164]. Another detected a suite of compounds with PSPT-like structures and activity, but the retention times of the compounds detected did not match standards, and therefore, TEFs could not be applied [4,60]. The three remaining studies could be used to estimate risk, but the PCOX studies tended to only detect STX and dcSTX, with only one reporting high concentrations of GTX4, which contrasts with the broader diversity in congener profiles reported in localized studies (Table 2). This could be a methodological PCOX artifact, possible chemical or biological instability of some PSPT congeners in the environment, or coincidence due to the limited number of studies. Regardless of the cause, it is important to further study PSPT profiles in a wide range of systems as this provides information for method selection.

The weight of evidence shows that PSPTs are common in freshwater environments, but the majority of the survey data cannot be reliably used to quantify human health risk. Future studies should measure a comprehensive panel of PSPTs, recognizing that there may not be standards available for many neurotoxic fractions, and report important test data (means, medians, ranges etc.), toxin concentrations before and after TEF conversion, and congener information (where appropriate). While ELISA is a useful screening tool to determine PSPT presence, additional methods should be included in future large-scale monitoring and surveillance studies.

In addition to environmental exposure, a novel route of toxin exposure that requires additional investigation are cyanobacterial dietary supplements, which have been demonstrated to contain microcystins [309]. A survey of spirulina powders available online suggest serving sizes of approximately ~3 g [310], although one product suggested consuming 7 tablets/day (>10 g/day). The presence of PSPTs in supplements has not been established, but contamination in STX eq. cannot exceed 11.67 µg STX eq./g supplement/day for a 70 kg adult (Box 1), assuming a NOAEL of 0.5 µg STX eq./kg b.w. While several studies have evaluated PSPTs in supplements, these studies have only targeted STX, and future measurements need to include additional congeners [310,311]. Preliminary PCOX screening analyses did not detect PSPTs in supplements evaluated by Miller et al., 2020 [309] (unpublished data).

## 8. Important Research Gaps for Freshwater Paralytic Shellfish Poisoning Toxins

### 8.1. Limited Availability and Type of Standards

Certified reference materials for some of the common marine PSPTs (~15 at the time of writing) are commercially available from the Canadian National Research Council. The majority of PSPT congeners are not available as certified materials or as purified, uncalibrated reference materials. Without reference standards, all LC-based methods may have difficulty with identification due to missing or uncertain retention time data, while it is even more difficult for MS detection methods, as mass spectrometers must be tuned to the appropriate ionization and collision conditions, which can be manufacturer and instrument dependent. This is due to the variability in the fragmentation patterns of the different PSPT congeners, with some congeners ionizing better in negative ionization mode. For MS methods, molecular weight can help identify compounds even when the exact retention time is uncertain, but oxidation-based methods do not have the same advantage. Many PSPTs have minimal functional group differences and therefore elute quite similarly in the Oshima [205] style solvent systems used in PCOX [4]. The use of ion-pair reagents hampers retention time predictions because it introduces another complex variable in the separation chemistry of the method. It can be difficult to estimate which PSPTs would be retained more (or less) when dealing with a complex mixture of PSPTs.

The toxicity, and thus the ability to estimate risk, from the majority of PSPTs is unknown or poorly evaluated due to the lack of purified material available. Assigning a TEF of 1 to congeners without adequate toxicity data may be protective, with the inbuilt assumption that this is likely to overestimate toxicity. However, there are some qualifiers to this, such as NEO recently having been re-evaluated for its oral toxicity, resulting in the TEF being increased to 2 [215]. The assumption that toxicity of PSPTs will always be less than STX could be a problematic assumption and put consumers at risk, especially considering layers of uncertainty thoroughly detailed throughout this review.

Addressing the lack of standards can be achieved in several ways. Total synthesis of STX [312] and derivatives [313,314] has been demonstrated, while synthesis of precursors by Japanese scientists has supported the chemical characterization of several new PSPT congeners in recent years [31,32,33]. Synthesis may not be economically feasible for all of the known PSPTs, but it could serve as an avenue to bring new reference materials to freshwater PSPT scientists. While some PSPTs are difficult to synthesize, some relatively simple reactions on purified PSPTs can produce new compounds, such as a sodium borohydride reduction of STX to produce either stereospecific 12β-do-STX at low temperatures or a mixture of 12α-do-STX and 12β-do-STX with higher temperatures [78,315].

The purification of STX and other derivatives from natural sources is where most commercially available reference materials are derived, either from cultured dinoflagellates or cyanobacteria or from shellfish homogenate. While the bulk of NRC Canada biotoxin reference materials are sourced from culture, some of the reference materials are synthetically produced. Purification is more affordable and can produce significant quantities of material. However, some of the “newer” PSPTs have been found in relatively low abundance [26], so there may be challenges in acquiring the quantities needed and at the needed purity. Recent studies have seen the transfer of components of the *sxt* gene cluster into *Escherichia coli* and the detection of some PSPT intermediates [316,317]. Patent WO 2017/137606 A1 demonstrates the recombinant production of NEO and other PSPTs by *E*. *coli* [318]. Other advancements include the transfer of *M. wollei* genes into *E. coli* and the isolation of Rieske oxygenase proteins, which were then used for the in vitro synthesis of several PSPT congeners and intermediates [319]. Further advancement in coming decades may lead to more readily available sources of PSPT reference materials.

### 8.2. Freshwater PSPT Congener Profiles and Environmental Drivers

The PSPT congeners produced by different cyanobacterial strains can vary widely (Table 2). This variability is due to inherent differences in the causative organisms, as well as the analytical methodologies implemented by researchers. Due to the importance of structure in determining the toxicity of PSPT congeners, it is essential that we better understand the identities of PSPTs present in freshwater systems. Benthic cyanobacteria may also produce a wholly different suite of PSPT congeners relative to their planktonic counterparts. Understanding the toxins present in the environment is critical to applying the optimal methods for their detection.

There is an assumption that “common” marine PSPTs are also important in freshwater systems. This assumption has stubbornly existed for decades, largely because of the crossover of marine PSPT congeners detected in freshwater systems and cyanobacterial strains. Critically, there has been an intense obsession with STX, specifically, as a freshwater PSPT congener, even though there has been no demonstration of this congener being the most abundant congener in freshwater PSPT producers. As STX and other common marine PSPTs are the best known, and for which there are existing standards, it makes sense that researchers would be more likely to detect these congeners compared to an unidentified PSPT with uncertain response factor in oxidation-based methodologies. However, it should not be assumed that the “old” PSPTs are the predominant or most important congeners for freshwater systems. Advancement in MS detection of PSPTs may begin to unravel this complexity due to its flexibility as a detector while also providing important compound-specific structural information.

PSPT profiles are influenced by the environment, including light, nutrient availability, and temperature [10], and there are also suggestions that there is a “circadian regulatory mechanism” involved in PSPT production [320,321]. Total nitrogen was highly correlated to PSPT concentrations in New York *Microseira wollei* with the growth of this cyanobacterium appearing to be nitrogen and light limited [12]. As an exceptionally nitrogen rich natural product, the regulation of PSPT metabolic pathways may be quite complex due to the quantity and types of nitrogen available to freshwater cyanobacteria. Establishing fundamental cellular regulation of PSPT production will make it easier to understand environmental shifts in PSPT profiles, which can have an outsized impact on toxin concentrations and relative toxicity.

### 8.3. Acute and Chronic Toxicity of PSPTs

While there is acute toxicological information for ~16 PSPT congeners [215,239] and several others that were published following the EFSA and FAO reports [267,322], there are dozens of PSPT congeners with much less or no information available regarding their toxicity. Since animal testing is expensive, new reference materials generated could be tested in vitro, as described in the FAO/WHO and EFSA PSPT reports, and in assays such as the RBA, to produce TEF estimates. As these data will take time to generate, the potential threat from these toxins should not be ignored because there are missing data but addressed with information that is available, even if it is imperfect. Importantly, it is crucial to account for these limitations and apply them to each individual study so that appropriate conclusions can be drawn.

The potential long-term health impacts from exposure to each of the PSPT congeners are poorly understood, yet it is imperative to study because of their presence in freshwater. The current paradigm has been the extrapolation of acute symptomology to determine PSPTs NOAELs to calculate drinking water standards, but these studies may need to consider additional endpoints that may be different for chronic exposure. The MBA is currently the “gold standard” method for PSPT analysis, with other methods referenced against the MBA to determine efficacy, but the MBA has severe limitations in freshwater due to its relatively high detection limit. A 2004 FAO report stated, “…that it is neither practical nor realistic to establish a very low tolerance level because the mouse bioassay is currently the most widely used method to determine PSP toxins and the present detection limit of this assay is approximately 40 µg PSP (STX eq)/100 g shellfish. Once more sensitive (and reliable) analytical chemical methods are available, the toxicity figures of STX and derivatives after acute and (sub)chronic exposure should be re-evaluated” [272]. Twenty years later, the limitations of the mouse bioassay (MBA) have not changed. There remain complications in establishing stronger safety measures because these safety measures are based on a lowest-observed-adverse-effect level (LOAEL) derived from human health events, many of which were quantified by the MBA. Using this LOAEL to calculate a freshwater regulatory limit may significantly underestimate the risks from intoxication due to the specific endpoints usually monitored in humans during an acute exposure event, while the testing methodology of the MBA itself is not especially sensitive and would not quantify at the concentrations of potential concern found in freshwater systems.

Most PSPT consumption guidelines do not include safety factors, with the only factor being a 3× division of the LOAEL to calculate a NOAEL. We believe safety factors are necessary to address the uncertainties in toxicological data. Animal testing would likely be needed for assessment of new congeners, acute versus chronic exposure scenarios, and/or the mode of action, which would introduce the need for one 10× safety factor. In addition to this safety factor for interspecies variation, introduced by using an animal model to measure toxic effects, others could be included for intraspecies variation and/or data uncertainties in the toxicological library. In the near term, the addition of safety factors to a NOAEL would mitigate exposure risk while additional toxicological data are generated. Assays and assessment of PSPT toxicity outside of their acute impact on the heart and lungs will be important, as there are reasons to believe there are deleterious impacts to other organs, including data showing the variability in time-to-death curves between congeners [266], their systemic distribution in the body [248], and activity to a range of ion channels [280].

## 9. Final Conclusions

PSPTs are a class of freshwater cyanobacterial toxins that have received little attention compared to their peers, such as microcystins. This is partially due to the difficulty of working with the PSPT class of molecules, which are difficult to isolate, analyze, and investigate for their biomolecular properties. However, there is enough evidence to state that identification of freshwater PSPTs is not an isolated geographic event but that they are produced worldwide by many cyanobacterial genera. The number of occurrences reported in this review is assuredly an underestimate, because each of the most common PSPT analytical and detection methods have intrinsic weaknesses that will produce false negatives. PSPT ELISAs are useful monitoring tools, but the data have sometimes been overinterpreted and conclusions have been drawn from data that cannot prove the original hypotheses. There should be far more caution when using and interpreting data from PSPT ELISA kit methods in the future. While PSPT presence has undoubtedly been established, the next crucial step for researchers is to better understand the risks associated with PSPTs. The scientific community needs a strong understanding of the biologic effects and toxicity of the suite of congeners, which is severely lacking for chronic and sub-chronic exposure. The availability and number of standards also needs to improve so that we can better use the analytical tools that do exist when searching for or quantifying environmental PSPTs. Freshwater PSPTs do share common features with their marine counterparts, but they should not be viewed or treated identically. If we are to effectively protect human health, a modern integrated approach will be required for the management of freshwater PSPTs.

## Figures and Tables

**Figure 1 marinedrugs-23-00271-f001:**
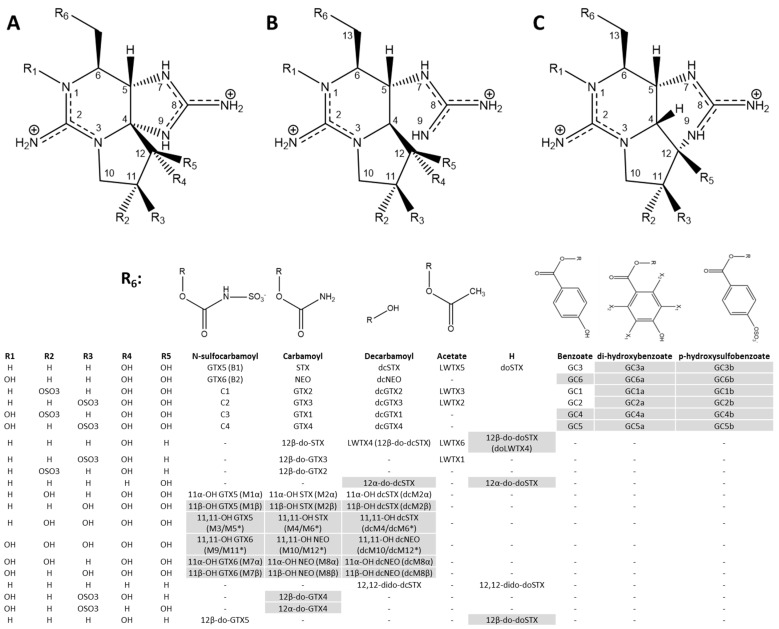
Structures of known PSPTs. The majority of PSPTs are associated with structure (**A**). Compounds in grey have been elucidated, but their structures are not fully validated by more exhaustive methods (NMR, synthesis, etc.). Compounds labeled with (*) represent M-toxin congeners where the C-N bond is broken between positions 4 and 9, structure (**B**), but have functional groups at positions on the standard saxitoxin tricyclic ring system. Two additional skeletal structures also exist associated with structure (**C**), which is the hemiacetal for the PSPTs M5-HA and M6-HA, and R groups corresponding with M5 and M6. Literature sources for each congener are in Table 1.

**Figure 2 marinedrugs-23-00271-f002:**
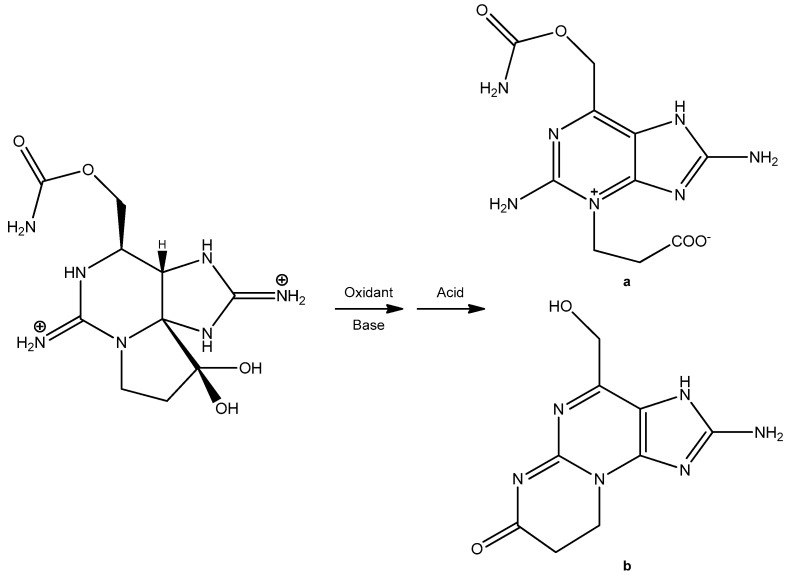
Scheme showing the conversion of saxitoxin to a fluorescent compound (**a**). Compound **a** converts to a second non-fluorescent compound (**b**) during isolation [40,208,210].

**Table 4 marinedrugs-23-00271-t004:** Relative response to STX for different analytical methods and assays.

Toxin	PCOX Relative Response [12]	MBA Relative Toxicity [215]	GSD/A STX-ELISA Cross-Reactivity ^a^ [109]	Receptor Binding Assay Relative Binding Affinity [191] ^†^	Receptor Binding Assay Relative Binding Affinity [216] ^†^
STX	1	1	1	1	1
NEO	0.41	0.5–1.2	0.013	0.73	3
GTX1	0.10	0.8–1	<0.02	1.04 **	0.47 **
GTX2	4.66	0.4	0.23	0.34 **	0.31 **
GTX3	3.52	0.6–1.1	0.23	0.34 **	0.31 **
GTX4	0.08	0.3–0.7	<0.02	1.04 **	0.47 **
GTX5	0.71	0.1–0.2	0.23	0.033	0.005
GTX6	0.44	0.1	-	-	0.017
dcSTX	1.13	0.4–1.02	0.29	0.10	0.53
dcNEO	0.30	0.02–0.4	0.06	-	0.24
dcGTX1	-	0.5	-	-	-
dcGTX2	2.71	0.2–0.3	0.014	-	0.075 **
dcGTX3	2.46	0.2–0.5	0.014	-	0.075 **
dcGTX4	-	0.5	-	-	-
LWTX1	0.09	0 [15]	0.13 *	-	-
LWTX2	-	0.11 [15]	0.13 *	-	-
LWTX3	-	0.06 [15]	0.13 *	-	-
LWTX4	-	0 [15]	0.13 *	-	-
LWTX5	-	0.14 [15]	0.13 *	-	-
LWTX6	-	0 [15]	0.13 *	-	-
C1+C2	1.20	0–0.2	-	-	0.007

^a^ Gold Standard Diagnostics/Abraxis (GSD/A) cross reactivities for GTX2,3, GTX1,4, dcGTX2,3 reported as epimeric mixtures for each pair of compounds. * Cross reactivity for all LWTXs reported as “Lyngbyatoxin”. Individual LWTXs are unlikely to have the same cross reactivity. ** Receptor binding activity for GTX2,3 and GTX1,4, dcGTX2,3 determined using epimeric mixtures. ^†^ RBA relative responses are reported as EC_50_ by Usup et al., 2004 [191] and Kd by Zakrzewska et al., 2025 [216].

**Table 5 marinedrugs-23-00271-t005:** Toxicity of different PSPT congeners relative to STX as measured by i.p. injections used in the MBA and their corresponding recommended TEFs [215].

Toxin	Relative Toxicity by MBA	FAO/WHO Toxicity Equivalency Factor
STX	1	1
NEO	0.5–1.2	2
GTX1	0.8–1	1
GTX2	0.4	0.4
GTX3	0.6–1.1	0.6
GTX4	0.3–0.7	0.7
GTX5	0.1–0.2	0.1
GTX6	0.1	0.05
dcSTX	0.4–1.02	0.5
dcNEO	0.02–0.4	0.2
dcGTX1	0.5	-
dcGTX2	0.2–0.3	0.2
dcGTX3	0.2–0.5	0.4
dcGTX4	0.5	-
LWTX1	0 [15]	-
LWTX2	0.11 [15]	-
LWTX3	0.06 [15]	-
LWTX4	0 [15]	-
LWTX5	0.14 [15]	-
LWTX6	0 [15]	-
C1+C2	0–0.2	-

**Table 6 marinedrugs-23-00271-t006:** An example of how variability in PSPT profile impacts an ELISA measurement compared against the toxicity equivalency factor (TEF) concentration of a set of PSPT samples. This is a thought experiment using reported cross-reactivity in the Gold Standard Diagnostics STX ELISA and the most up-to-date TEFs reported for PSPTs. STX, saxitoxin; GTX2,3, gonyautoxin2,3; GTX1,4, gonyautoxin1,4.

	[STX]	[GTX2,3]	[GTX1,4]	ELISA Result *	True TEF Adjusted Toxin Concentration	ELISA % Error ^†^
Sample A	1	1	0	1.23	1.50	18.00%
Sample B	1.5	0.5	0	1.62	1.75	7.43%
Sample C	0.5	1.5	0	0.84	1.25	32.40%
Sample D	10	1	0	10.23	10.50	2.57%
Sample E	1	10	0	3.30	6.00	45.00%
Sample F	1	1	1	1.25	2.35	46.81%
Sample G	1	1	10	1.43	10.00	85.70%

GTX isomers are assumed to be split 50:50 in this example. The TEFs were taken from the WHO/FAO report (Table 5) [215]. * The ELISA result is calculated using the cross-reactivities reported in the Gold Standard Diagnostics/Abraxis ELISA (Table 4) [109]. ^†^ ELISA percent error is represented as the percent difference from the concentration of the ELISA result to the concentration after TEF adjustment.

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
