# Peer review of "A Fresh Perspective on Cyanobacterial Paralytic Shellfish Poisoning Toxins: History, Methodology, and Toxicology"

_marinedrugs, 2025, doi:10.3390/md23070271_

Round 1
Reviewer 1 Report
Comments and Suggestions for Authors
This review was extensive and detailed.
I have several comments and suggestions to address.
Lines 352-365 – the reporting of naturally fluorescent compounds in bacterial cultures, which were mistaken as PSTS, occurred in some early HPLC studies from the 1990s. The authors did not approach this aspect at this point, but only much later (section 5.2.3)
Section 4.1 - The authors did not talk about any special requirements to deal with the radioactive isotopes used in the RBA method. This might be a drawback for the use of this method in several laboratories.
Table 4: Is reference nº 4 the one adequate here for the PCOX? In line 587, it was used reference nº 11 for discussing the relative responses upon oxidation.
Line 640- ‘more’ is repeated twice in the same sentence.
Section 5.1. Mass spectrometry detection is discussed as a targeted method. However, complementing with fluorescence might be important for novel strains or water bodies that have not been studied before. Due to the large number of PSTS, it is not practical to have a single MS method which screens all the variants. The screening of multiple compounds decreases method sensitivity. Some variants or novel toxins might be missed with a targeted method.
Line 808-820- The pre-column oxidation method was not discussed here. It has the advantage of looking for naturally fluorescent compounds without traces of the oxidant in the system. It can also be quite sensitive.
Line 906-907- «concentrations of afflicted individuals» is not correct. Toxin concentrations are measured in food, not in individuals. Toxins in individuals are estimated from food ingestion.
Line 938-939- This article is rather long. However, regarding toxin concentrations naturally occurring in freshwater and drinking water, there is little mention here. The data presented earlier (Table 3), was not discussed at this point.
Table 6- The caption for this table does not specify that these samples were made from pure standards, weren’t they? The same comment for the text below.
Line 1254- standards from the CIFGA company include additionally C3+4 toxins.
In this review, there was no mention of freshwater toxins present as ‘food’, with the widespread popularization of algae supplements (such as Spirulina, and others).
Author Response
Please see attachment. We have attached our response to each set of reviewer comments for your benefit.

Reviewer 2 Report
Comments and Suggestions for Authors
This is a very thorough review of the literature on paralytic shellfish poisoning toxins in marine and freshwater environments. A wide range of topics especially related to freshwater PSPTs, from histry to chronic toxicity, are discussed in a comprehensive manner. Furthermore, future challenges regarding freshwater PSPT were highlighted. It is an important contribution to the field, and should be published. However, there are numerous points that need to be revised. Therefore, I recommend reconsider after major revisions.

Author Response

(The authors gave the same response as above.)

Reviewer 3 Report
Comments and Suggestions for Authors
The review is clear and exhaustive, describing in depth the issue of Paralytic Shellfish Toxins (PSPTs) in freshwater systems, surely an emerging concern in this field. These toxins are a potent neurotoxin class, well known and investigated in marine ecosystems, but there is few information about their presence in freshwater environments. The aim of the review is to give an overview on PSPT presence in freshwater systems with toxic profiles, analytical methods available for the detection and information about toxicology and potential risks for human health.
A complete list of PSPTs so far known is shown, including also toxins for which structures are only elucidated but not fully validated. A detailed description of PSPTs detected in freshwater systems with location, source, detection method and toxic profile was reported, giving an exhaustive history. Moreover an overview on detection methods actually available was done, from biochemical assay to instrumental ones such as HPLC coupled to fluorescence and mass spectrometry. The methods described are applied in shellfish and marine ecosystems, but poorly investigated on freshwater systems. The main challenges are the presence of new PSPT analogues and in a wide variety of matrices, the lack of standards and poor understanding on toxic profile.
Toxicology and human health concerns were also investigated underling the lack of data, in particular related to chronic and sub-chronic exposure. In fact in freshwater environments, the main source of exposure to PSPTs is represented from drinking and recreational waters. So far, the data available are limited to oral toxicity tested by mouse bioassay.
In conclusion the review showed that PSPTs are present in freshwater systems, produced worldwide by many cyanobacterial genera. Therefore to protect human health, a modern integrated approach will be required for the management of this potential threat.
Some sections such as “Methods for Detection”, “Toxicology and Human Concerns from Freshwater PSPTs “ should be synthetize because although providing an accurate overview, some information reported is not relevant to the purpose of the review.
Moreover in the section “Selecting an appropriate method for freshwater PSPT analysis” the potential of high resolution mass spectrometry, to perform untargeted analysis for unknown PSPTs, should be emphasize.
The superscript “*” should be replace with “1” as reported in the text (line 37) and the motivation for the choice of acronym synthesized.
In the bibliography the references regarding the AOAC methods such as “AOAC 959.08”, “ AOAC 2005.06”, “ AOAC 2011.02” should be added.
Specific comments are reported in the attached file.

Author Response

(The authors gave the same response as above.)

Round 2
Reviewer 1 Report
Comments and Suggestions for Authors
The authors followed adequately the reviewer's sugestions and improved the ms.
Author Response
Thank you very much for reviewing our manuscript for a second time. We greatly appreciate your time.
Reviewer 2 Report
Comments and Suggestions for Authors
The manuscript has been revised quite well, and the letter answered the questions well, but there are still some critical errors that need to be corrected.
Figure 1
The structure of GTX2, GTX3, dcGTXs, dcGTX3, GTX1, GTX4, dcGTX1, dcGTX4, 12ß-do-GTX2, 12ß -do-GTX3, 11α-OH dcSTX, 11ß-OH STX, 12ß-do-GTX4, 12α-do-GTX4 are incorrect.
The stereochemistry at the position of 11 are opposite for GTX2/GTX3, GTX1/GTX4 and corresponding dc-type.
11ß-OH STX (dcM2 ß) should be 11ß-OH dcSTX (dcM2 ß).
12ß-do-GTX4 and 12α-do-GTX4 should be 12ß-do-GTX1 and 12α-do-GTX1.
Table 1
If the charge is 1, the superscript number is not necessary in elemental formula.
Line 241-250
In ref#57, I could not find the evidence of doGTX2/3. In Table 1 of #57, “doGTX2/4” are listed but not doGTX2/3. They only described “deoxy-decarbamoyl derivative of gonyautoxin (doGTX2/3)” in the text. We cannot know which is true. Please delete “but this pair ---C-13 in line 242 -244.
Line 246
fromsm ?
Line 327 – 329
“The carbamoyl functional group has been found in several forms” is not correct.
Please rephrase this part as it means that the structure of functional groups at the position of C13.
Line 334
“dicarbamoyl” must be “decarbamoyl”
Line 1535
Ref 314 is not the only article which used STX biosynthetic genes. The works by Dr. Narayan’s group should also be referred to in this context.
“SxtA” should be “sxtA”, the first letter should not be capitalized.
Ref#32
Hakamada, M.; Tokairin, C.; Ishizuka, H.; Adachi, K.; Osawa, T.; Aonuma, S.; Hirozumi, R.; Tsuchiya, S.; Cho, Y.; Kudo, Y. Synthesis and Identification of decarbamoyloxySaxitoxins in Toxic Microalgae and Their Reactions with the Oxygenase, SxtT, Reveal Saxitoxin Biosynthesis. Chemistry–A European Journal 2024, 30, 1–9, doi:https://doi.org/10.1002/chem.202304238.
is incorrect.
4 people are missing. Keiichi Konoki , Yasukatsu Oshima , Kazuo Nagasawa , Mari Yotsu-Yamashita
Please check it again.
Compounds name
There are spaces between GTX and the number in this manuscript, however usually compound’s names are written without space like GTX5. The names in Figure 1 are correct. Please check the names in the text, Tables and legends and correct.
Author Response
Please see the attached word document

Round 3
Reviewer 2 Report
Comments and Suggestions for Authors
The manuscript has been well corrected, so I think it is ready for publication.